# *Tgfbr1* regulates lateral plate mesoderm and endoderm reorganization during the trunk to tail transition

**Anastasiia Lozovska**[1†], **Ana Casaca**[1,2], **Ana Novoa**[1,2], **Ying-Yi Kuo**[3], **Arnon D Jurberg**[1‡], **Gabriel G Martins**[1,2], **Anna-Katerina Hadjantonakis**[3], **Moises Mallo**[1,2]*

[1]Instituto Gulbenkian de Ciência, Rua da Quinta Grande, Oeiras, Portugal; [2]Gulbenkian Institute for Molecular Medicine, Avenida Prof. Egas Moniz, Lisboa, Portugal; [3]Developmental Biology Program, Sloan Kettering Institute, Memorial Sloan Kettering Cancer Center, New York, United States

**\*For correspondence:**
moises.mallo@gimm.pt

**Present address:** †Department of Molecular Genetics and Microbiology, University of Florida, Gainesville, United States; ‡Universidade Estácio de Sá (UNESA)/Instituto de Educação Médica (IDOMED) - Campus Vista Carioca, 20071-004 Rio de Janeiro/RJ, Brazil, and Laboratório de Animais Transgênicos, Universidade Federal do Rio de Janeiro (UFRJ), Rio de Janeiro, Brazil

**Competing interest:** The authors declare that no competing interests exist.

## eLife Assessment

Morphological characteristics and phenotypes of mutations in key developmental genes suggest that head, trunk, and tail development are regulated by discernible modules. Gdf11 signalling plays a crucial role in orchestrating the transition from trunk to tail tissues in vertebrate embryos. This **important** study presents **convincing** evidence that Tgfbr1 acts upstream of Isl1 (a pivotal effector of Gdf11 signalling) and regulates blood vessels, the lateral plate mesoderm, and the endoderm associated with the trunk-to-tail transition. Together with the previous studies, this work identifies a key signal that acts as the pivot of the trunk-to-tail transition.

**Abstract** During the trunk to tail transition the mammalian embryo builds the outlets for the intestinal and urogenital tracts, lays down the primordia for the hindlimb and external genitalia, and switches from the epiblast/primitive streak (PS) to the tail bud as the driver of axial extension. Genetic and molecular data indicate that Tgfbr1 is a key regulator of the trunk to tail transition. Tgfbr1 has been shown to control the switch of the neuromesodermal competent cells from the epiblast to the chordoneural hinge to generate the tail bud. We now show that in mouse embryos Tgfbr1 signaling also controls the remodeling of the lateral plate mesoderm (LPM) and of the embryonic endoderm associated with the trunk to tail transition. In the absence of Tgfbr1, the two LPM layers do not converge at the end of the trunk, extending instead as separate layers until the caudal embryonic extremity, and failing to activate markers of primordia for the hindlimb and external genitalia. The vascular remodeling involving the dorsal aorta and the umbilical artery leading to the connection between embryonic and extraembryonic circulation was also affected in the Tgfbr1 mutant embryos. Similar alterations in the LPM and vascular system were also observed in Isl1 null mutants, indicating that this factor acts in the regulatory cascade downstream of Tgfbr1 in LPM-derived tissues. In addition, in the absence of Tgfbr1 the embryonic endoderm fails to expand to form the endodermal cloaca and to extend posteriorly to generate the tail gut. We present evidence suggesting that the remodeling activity of Tgfbr1 in the LPM and endoderm results from the control of the posterior PS fate after its regression during the trunk to tail transition. Our data, together with previously reported observations, place Tgfbr1 at the top of the regulatory processes controlling the trunk to tail transition.

## Introduction

The transition from trunk to tail development is a complex process resulting in major changes in the general structure of the embryo, also involving a switch in the mechanisms regulating axial extension. Extension through the trunk is driven by axial progenitors located within the epiblast, that generate the spinal cord, the embryonic gut, and the different mesodermal compartments (*Binagui-Casas et al., 2021*; *Cambray and Wilson, 2007*; *Henrique et al., 2015*; *Steventon and Martinez Arias, 2017*; *Tsakiridis and Wilson, 2015*; *Wilson et al., 2009*; *Wymeersch et al., 2021*). At this stage, the caudal end of the mouse embryo is occupied by the allantois that will play an essential role in the connection between embryonic and extraembryonic structures (*Arora and Papaioannou, 2012*; *Rodriguez and Downs, 2017*). The transition to tail development is associated with changes in the global anatomy of the caudal end of the embryo, involving the progressive anterior relocation of the allantois along the ventral side of the embryo. During this process, the tail bud forms at the dorsal and posterior end of the embryo, and replaces the epiblast/primitive streak (PS) as the main driver of axial extension (*Henrique et al., 2015*; *Wilson et al., 2009*). Formation of the tail bud results from changes in the progenitors generating the neural and paraxial mesodermal structures, the so-called neuro-mesodermal-competent (NMC) population, which relocates from the epiblast to the chordo-neural hinge (CNH) (*Binagui-Casas et al., 2021*; *Cambray and Wilson, 2007*; *Wymeersch et al., 2021*).

At this stage, the lateral plate mesoderm (LPM) also undergoes a major reorganization. This meso-dermal compartment, generated by progenitors situated at the caudal region of the epiblast and PS (*Wymeersch et al., 2021*; *Wymeersch et al., 2019*; *Wymeersch et al., 2016*) is composed of two layers: a ventral splanchnopleure, which contributes to the formation of the various body organs, as well as their vascularization, and a lateral somatopleure involved in the formation of the body wall (*Prummel et al., 2020*). These two layers delimit the celomic cavity, which will hold the animal's internal organs. During allantois relocation, the two LPM layers converge toward the midline, ending the celomic cavity and marking the posterior border of the trunk. This remodeling of the caudal part of the embryo is associated with the induction of the hindlimbs from the somatopleure (*Tickle, 2015*), and the generation of the pericloacal mesenchyme, the primordium of the genital tuberculum (GT) (*Cohn, 2011*; *Yamada et al., 2006*), from the ventral lateral mesoderm (VLM) posterior to the allantois (*Tschopp et al., 2014*).

Concomitant with the reorganization of the embryonic mesoderm, the transition from trunk to tail development also involves major changes in the embryonic endoderm and in the vascularization that will connect embryonic and extraembryonic structures. When the allantois relocates, the embryonic endoderm, whose posterior end reaches the base of the allantois, forms a cavity that will originate the cloaca, an endodermal expansion that becomes the common end of the excretory, intestinal, and genital tracts (*Huang et al., 2016*; *Matsumaru et al., 2015*). The pronephric ducts, derivatives of the intermediate mesoderm (IM) located medially to the splanchnic and somatic LPM layers, later merge with the cloaca to engage in the development of the urogenital system (*Davidson, 2008*). In the mouse embryo, the embryonic endoderm then expands further caudally to generate the tail gut, a transient structure with unknown function. The region of the posterior visceral (extraembryonic) endoderm abutting the allantois is thought to facilitate the invagination and growth of the embryonic endoderm (*Rodriguez and Downs, 2017*) and contribute to the hindgut epithelium (*Kwon et al., 2008*; *Nowotschin et al., 2019*).

The major blood vessels also become reorganized with the relocation of the allantois. The caudal end of the paired dorsal aortae (DA) merge and connect with the umbilical artery generated within the allantois (*Arora and Papaioannou, 2012*; *Downs and Rodriguez, 2020*). As the allantois move forward, the caudal end of the DA bends to form the recurved dorsal aorta (rDA). It is thought that this process requires the generation of a vessel of confluence from the caudal end of the PS abutting the allantois, which will constitute a major part of the rDA (*Downs and Rodriguez, 2020*; *Rodriguez and Downs, 2017*). Reorganization of the DA/umbilical artery connection will generate the blood vessels linking the embryo with the placenta and will also generate the arteries that will provide irrigation to the hindlimbs.

Molecular and genetic data indicate that Gdf11 signaling is an integral component of the gene regulatory network controlling the trunk to tail transition (*Aires et al., 2019*; *Jurberg et al., 2013*; *Matsubara et al., 2017*; *McPherron et al., 2009*; *McPherron et al., 1999*). Gdf11 activity is predom-inantly mediated by transforming grow factorβ receptor 1 (Tgfbr1) (also known as Alk5) (*Andersson*

*et al., 2006*). Indeed, premature expression of a constitutively active form of Tgfbr1 promotes early execution of the trunk to tail transition program (*Jurberg et al., 2013*). In addition, *Tgfbr1* is required to trigger formation of the tail bud by promoting, together with *Snai1*, an incomplete epithelial-to-mesenchymal transformation (EMT) in the NMC population within the epiblast (*Dias et al., 2020*). In the present study, we show that, in addition to the lack of molecular signals for the induction of the hindlimbs and the GT, the LPM of *Tgfbr1* null mutants fail to converge leading to the posterior extension of the celomic cavity. In addition, the mutants fail to generate a cloacal cavity and to extend the endodermal tube to form the tail gut. Also, the connection between the embryonic and extraembryonic vascular systems fails to undergo normal reorganization, resulting in the expansion of the paired dorsal aorta to reach the caudal end of the embryo. We also provide evidence indicating that *Isl1* is the key functional downstream target of *Tgfbr1* for the reorganization of the LPM and vascular tissues during the trunk to tail transition, acting on the posterior PS/allantois. *Isl1* controls PS fate during its regression and regulates the establishment of the embryonic–extraembryonic connection during the ventral relocation of the allantois. Taken together, our findings indicate that *Tgfbr1* is a master regulator of the trunk to tail transition.

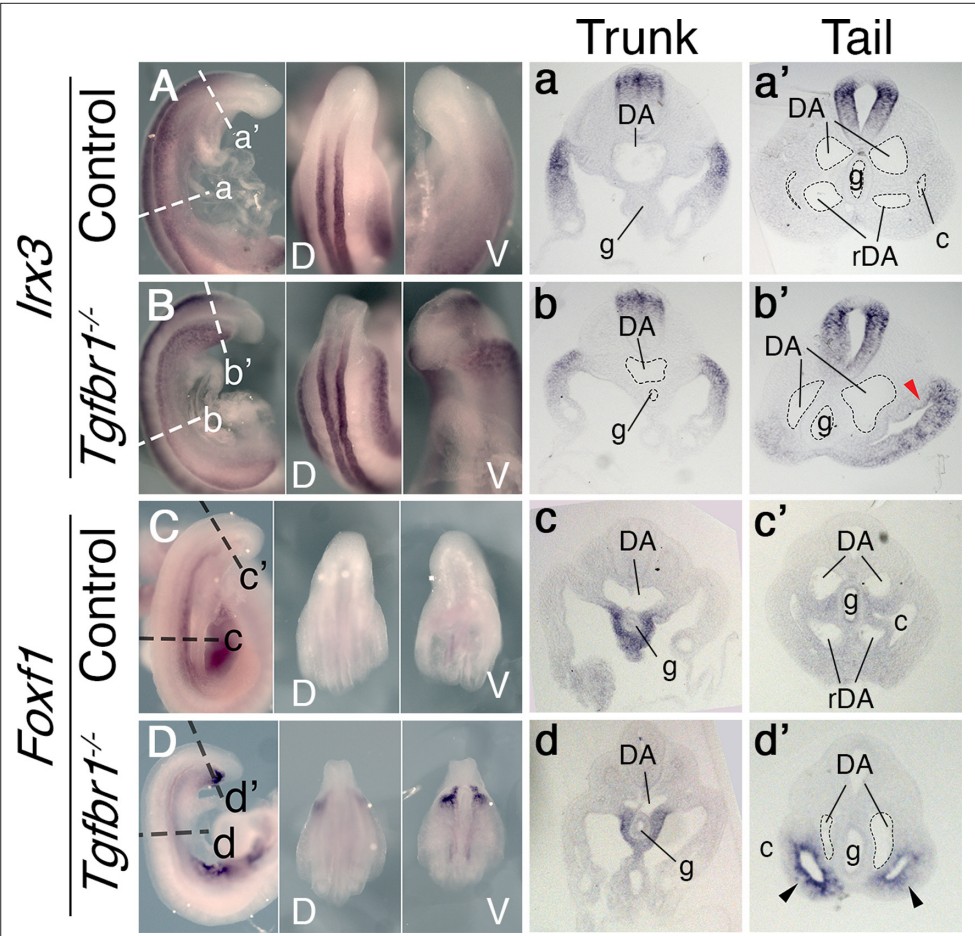

**Figure 1.** In situ hybridization showing expression patterns of the main mesodermal markers. (**A–d'**) Expression of somatic lateral plate mesoderm (LPM) marker Irx3 (**A, B**) and splanchnic LPM marker Foxf1 (**C, D**) in control (**A, C**) and Tgfbr1⁻/⁻ (**B, D**) E9.5 embryos. Next to the images of the whole-mount embryos shown transversal sections through trunk (a–d) and tail (a'–d') regions. Red arrowhead indicates ectopic expression of Irx3 in splanchnic LPM, black arrowhead – ectopic expression of Foxf1 in somatic LPM. c – coelomic cavity, cl – cloaca, DA – dorsal aorta, g – gut, rDA – recurved dorsal aorta, V – ventral, D – dorsal.

The online version of this article includes the following figure supplement(s) for figure 1:

**Figure supplement 1.** Expression of the intermediate mesoderm (IM) marker *Pax2* in *Tgdfr1⁻/⁻* embryos.

## Results

### *Tgfbr1* is a key modulator of the caudal trunk mesoderm differentiation

We previously showed that *Tgfbr1* null mutant embryos fail to activate markers labeling the hindlimb and GT primordia (the VLM) (*Dias et al., 2020*). We now assessed the defects of the *Tgfbr1* mutants at the axial level at which these genes become active in wild-type embryos. Transverse sections through this area indicated abnormal morphology of the LPM (*Figure 1*). In wild-type embryos, the somatic and splanchnic LPM layers converge at the posterior region of the trunk at both sides of the developing endoderm at E9.5, ending the celomic cavity. In *Tgfbr1* mutant embryos, however, the two LPM layers, and, consequently, the coelomic cavity, continued extending from the trunk region of the embryo until its posterior end (*Figure 1b', d'*). Despite this apparent extension of the trunk LPM into the prospective hindlimb and VLM regions, molecular analyses suggested that the LPM properties in this area differed from those observed in the trunk. This was best illustrated by the expression of *Irx3* and *Foxf1*, somatic and splanchnic LPM markers, respectively (*Funayama et al., 1999*; *Mahlapuu et al., 2001*). Both markers were expressed following normal patterns in the trunk of *Tgfbr1⁻/⁻* embryos (*Figure 1Aa, Bb, Cc, Dd*). However, in contrast to wild-type controls, in *Tgfbr1* null mutant embryos these markers were not downregulated at the level of the trunk to tail transition, being *Foxf1* even clearly upregulated in this embryonic region (*Figure 1B, D*). In addition, their expression was no longer restricted to their respective LPM layers, and instead expanded to encompass the entire mesodermal tissue surrounding the celomic cavity. This feature was more clearly observed for *Foxf1* (*Figure 1Dd'*). The expression pattern of the IM marker *Pax2* at E10.5 revealed that the pronephric ducts also extended into the posterior end of the embryo instead of merging with the cloaca (*Figure 1—figure supplement 1c, c' , d"*). Furthermore, pronephric ducts were bifurcated in the posterior embryonic end in some mutant embryos (2/4) (*Figure 1—figure supplement 1D-d"*).

Another major mesodermal derivative affected by the absence of *Tgfbr1* was the main vascular tree, particularly the region connecting embryonic and extraembryonic circulation. The four orifices surrounding the hindgut in E9.5 embryos observed in transverse sections, and diagnostic of the rDA (*Zakin et al., 2005*), were not detected in *Tgfbr1* mutants, in which only a single expanded vessel was visible on each side of the gut tube (*Figure 1Aa', Bb', Cc', Dd'*). Pecam1-aided 3D reconstruction of the main blood vessels revealed that the DA of *Tgfbr1* mutant embryos was elongated posteriorly, reaching the tip of the gut tube. The posterior portion of the DA formed a vessel of enlarged diameter, as observed in the histological sections, that contrasts with the curvature characteristic of the rDA of wild-type embryos (*Figure 2*). The DA of the *Tgfbr1* mutants still merged with the umbilical artery, although following patterns different to those observed in control embryos. For instance, while in wild-type embryos the paired rDAs remained as two independent vessels, merging just before their connection with the umbilical artery (*Figure 2b*), the two paired arteries of *Tgfbr1⁻/⁻* embryos were fused along most of their path ventral to the endodermal tube, from the posterior embryonic tip to the umbilical artery. In addition, the allantois appeared to protrude perpendicularly to the embryo instead of following its curvature, as observed in wild-type embryos (*Figure 2d*).

Together, the above observations indicate that in the absence of *Tgfbr1* the mesodermal tissues derived from the lateral mesoderm fail to execute the normal differentiation routes associated with the trunk to tail transition.

### Possible involvement of the posterior PS as mediator of Tgfbr1 activity

Splanchnic *Foxf1* expression depends on endodermal Shh activity (*Astorga and Carlsson, 2007*; *Tsiairis and McMahon, 2009*). The *Foxf1* expression observed in the lateral layer of the expanded LPM of *Tgfbr1* mutant embryos is separated from the endoderm by the celomic cavity, suggesting that it is likely to have a Shh-independent origin. Such *Foxf1* expression has been detected in the posterior PS/allantois of E8.5 embryos (*Astorga and Carlsson, 2007*; *Tsiairis and McMahon, 2009*), where it plays a role in vasculogenesis (*Astorga and Carlsson, 2007*). Reorganization of the vascular system to connect the embryonic and extraembryonic circulation occurs within the emergent VLM as the allantois becomes displaced anteriorly during the trunk to tail transition (*Downs and Rodriguez, 2020*; *Rodriguez and Downs, 2017*). From a functional perspective, *Foxf1* expression in the VLM is likely to derive from its expression domain in the posterior PS/allantois (*Figure 2—figure supplement 1*). *Foxf1* expression in the posterior PS/allantois is not affected by the absence of *Tgfbr1* (*Figure 3A, B*), consistent with the apparent dispensability of *Tgfbr1* activity in axial tissues before

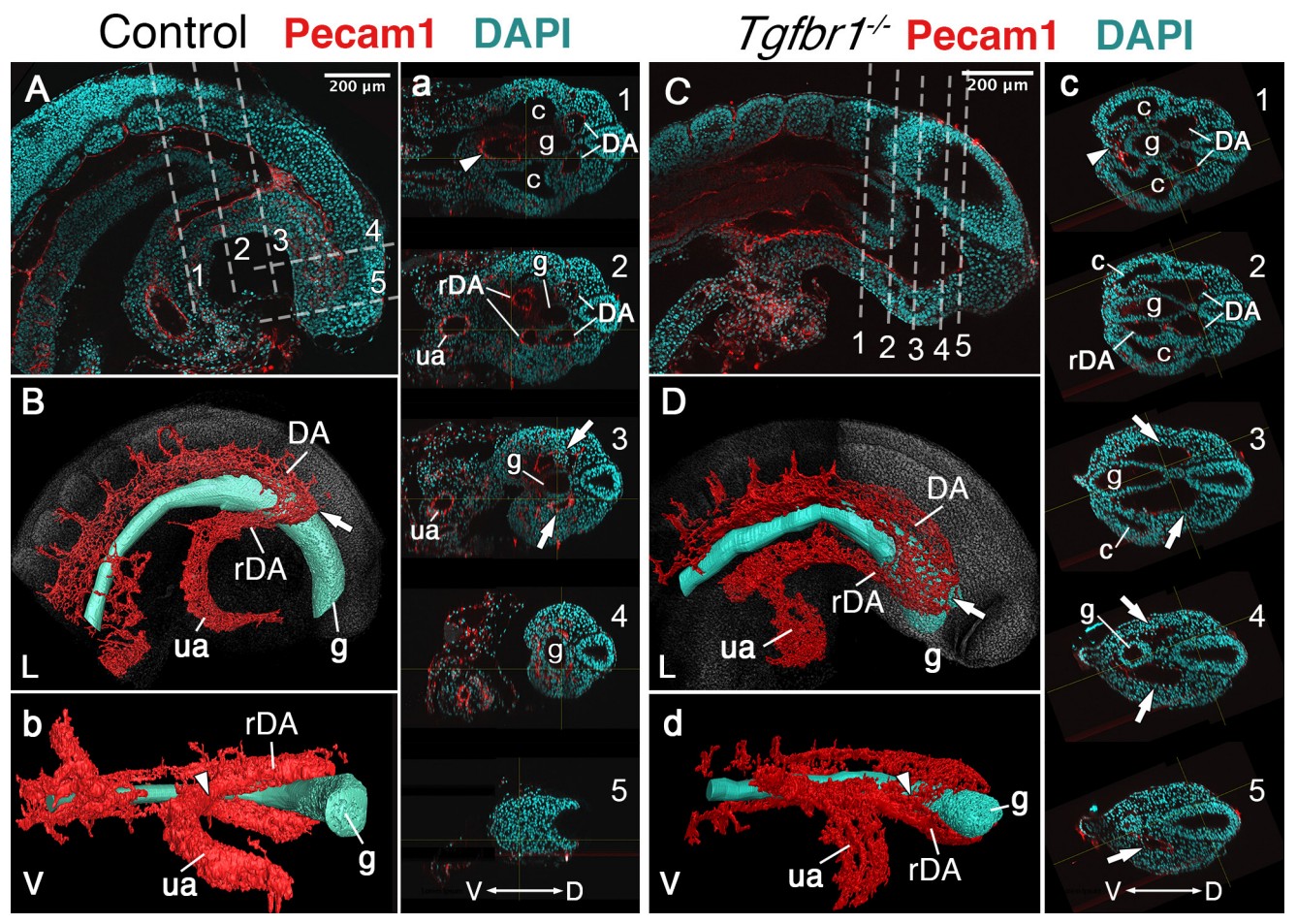

**Figure 2.** Main vascular tree of the Tgfr1⁻ᐟ⁻ embryos. Whole-mount immunostaining for Pecam1 (red) labeling endothelial cells in E9.5 control (**A, a**) and mutant (**C, c**) embryos. Nuclei shown in cyan. Transversal sections through regions marked by the dashed lines in A and C are shown in (a1–5) and (c1–5). (**B, b, D, d**) 3D reconstruction of the main vascular tree (red) and the gut (cyan) of the immunostaining shown in (**A, a, C, c**). Connection between the umbilical artery (ua) and recurved dorsal aortae (rDA) is marked by the arrowhead. Turn of dorsal aortae (DA) where it is connected to rDA is labeled by the arrow. In the mutant this region is enlarged while rDA is short (compare **A**, a3, and **B** to **C**, c2–5, and **D**). D – dorsal, L – lateral, V – ventral, c – coelomic cavity, g – gut.

The online version of this article includes the following figure supplement(s) for figure 2:

**Figure supplement 1.** Pericloacal mesenchyme derives from the mesoderm adjacent to the allantois.

the transition to tail development. Given the absence of VLM in *Tgfbr1⁻ᐟ⁻* embryos, the *Foxf1*-positive cells in the extended LPM might represent the derivatives of posterior PS/allantois cells that failed to enter their normal fates, which instead become intermingled with LPM cells extending from the trunk. Indeed, the abnormal development of the posterior aortae in *Tgfbr1* mutant embryos might result from compromised development of the *Foxf1*-positive posterior PS/allantois.

To assess whether the PS contributes to the prospective pericloacal region, we generated a transgenic line (*T-str-creERT*) expressing the tamoxifen-inducible cre recombinase in the PS under the control of an enhancer of *Brachyury* (currently known as *Tbxt*) (*Clements et al., 1996*) and inducing cre-mediated ROSA26-derived reporter expression (*Soriano, 1999*; *Srinivas et al., 2001*; *Figure 3—figure supplement 1*). Tamoxifen administration at E8.0, which activates cre activity within the 10–12 hr corresponding to E8.5 (*Figure 3—figure supplement 2*), thus coincident with the start of the trunk to tail transition, resulted in labeling of the VLM of E9.5 embryos, and of the pericloacal mesoderm at E10.0 (*Figure 3c*). Although this experiment does not allow regional sub localization within the PS, it is consistent with the *Foxf1*-positive posterior PS/allantois being a functional target of *Tgfbr1* activity in the LPM during the trunk to tail transition.

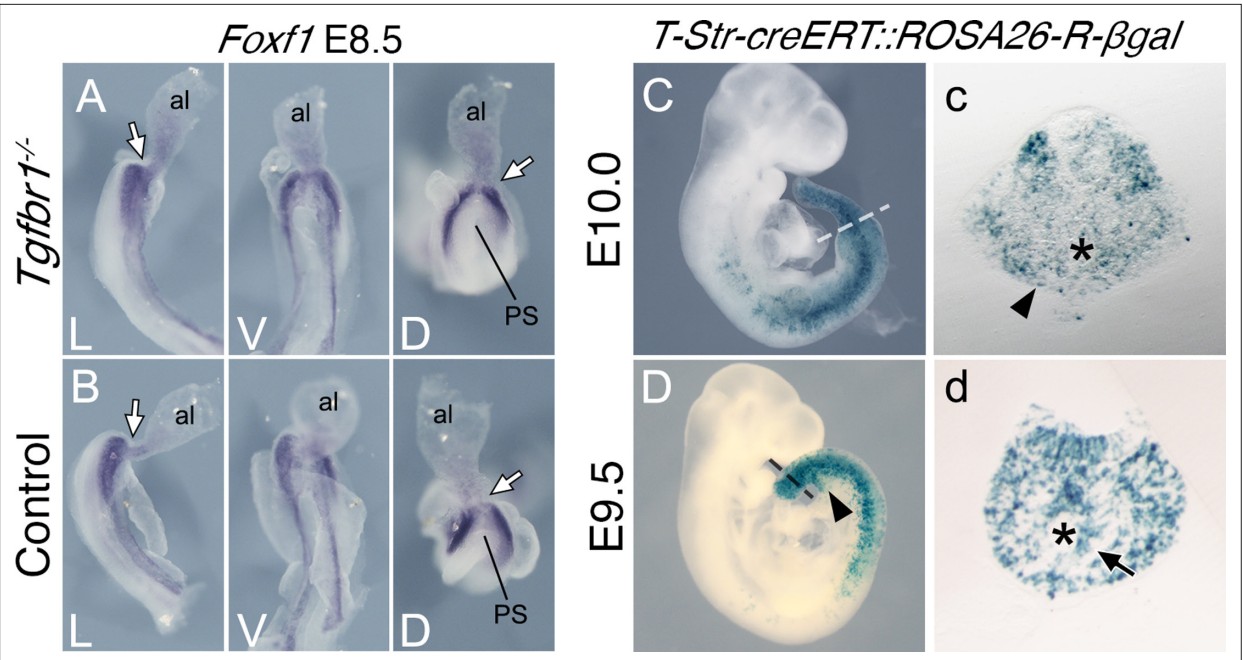

**Figure 3.** Posterior primitive streak contributes to the pericloacal mesenchyme and gut endoderm. Whole-mount in situ hybridization showing expression of *Foxf1* in E8.5 *Tgfbr1*⁻/⁻ (**A**) and control (**B**) embryos. al – allantois, g – gut, PS – primitive streak, L – lateral view, V – ventral view, D – dorsal view. While arrow indicates PS/allantois junction. β-Galactosidase cell tracing showing descendance of the primitive streak in the E10.0 (**C, c**) and E9.5 (**D, d**) embryos. c and d show transversal sections through regions marked by the dashed lines in C and D. Black arrowhead shows β-galactosidase staining in the pericloacal mesenchyme. Black arrow in d shows β-galactosidase⁺ cells in the tail gut endoderm. The asterisk in c indicates the cloaca. The asterisk in d indicates the tail gut.

The online version of this article includes the following figure supplement(s) for figure 3:

**Figure supplement 1.** Characterization of the recombination activity of the *Tstr-cre*ᴱᴿᵀ transgenics.

**Figure supplement 2.** Estimating the time required for recombination after tamoxifen administration in *Tstr-cre*ᴱᴿᵀ::*ROSA26R-YFP* embryos.

## *Isl1* mediates *Tgfbr1* activity in the lateral mesoderm

Reporter, gain and loss of function experiments suggested that *Isl1* might be functioning downstream of Tgfbr1 signaling in the LPM during the trunk to tail transition (*Dias et al., 2020*; *Jurberg et al., 2013*). It has been shown that *Isl1* first becomes activated in axial tissues at the PS/allantois junction, just prior to the transition to tail development (*Cai et al., 2003*; *Wymeersch et al., 2019*), and Isl1-positive cells later contribute to the VLM, the hindlimbs and GT (*Yang et al., 2006*). Analysis of the role of *Isl1* during the trunk to tail transition might thus provide an independent test of the functional relevance of the posterior PS/allantois for *Tgfbr1* activity in the LPM during the transition. We generated *Isl1* null mutant embryos using the null allele resulting from the insertion of the cre recombinase replacing the *Isl1* gene in the *Isl1-cre* strain (*Srinivas et al., 2001*). *Isl1* null embryos were embryonic lethal between E9.5 and E10.5, showing malformations in different embryonic structures. Regarding axial development, *Isl1* mutants halted their growth around the stage of the trunk to tail transition, as estimated by the number of somites generated (*Figure 4D–F, I, J*), thus reminiscent of the *Tgfbr1* mutant phenotype. However, in contrast to *Tgfbr1* mutants, *Isl1* mutants generated a structure resembling the tail bud which expressed *Sox2* and *Tbxt* in domains comparable to wild-type embryos (*Figure 4A, B, D, E*). The presence of a tail bud in *Isl1* mutants indicates that this gene might not be involved in the activity of the NMC cells, as suggested by its expression pattern (*Cai et al., 2003*). However, we observed major morphological and molecular alterations in the region corresponding to the LPM. Interestingly, some of those alterations are comparable to the defects observed in *Tgfbr1* mutants. The two layers of the lateral mesoderm, as well as the celomic cavity, extended to the posterior extremity of the embryo (*Figure 4H, h', h''*). *Foxf1* expression was also upregulated in the posterior of the embryo, showing a spatial distribution that starts at the dorsal border between the splanchnic and somatic LPM layers, extending to fully cover the somatic lateral mesoderm at more

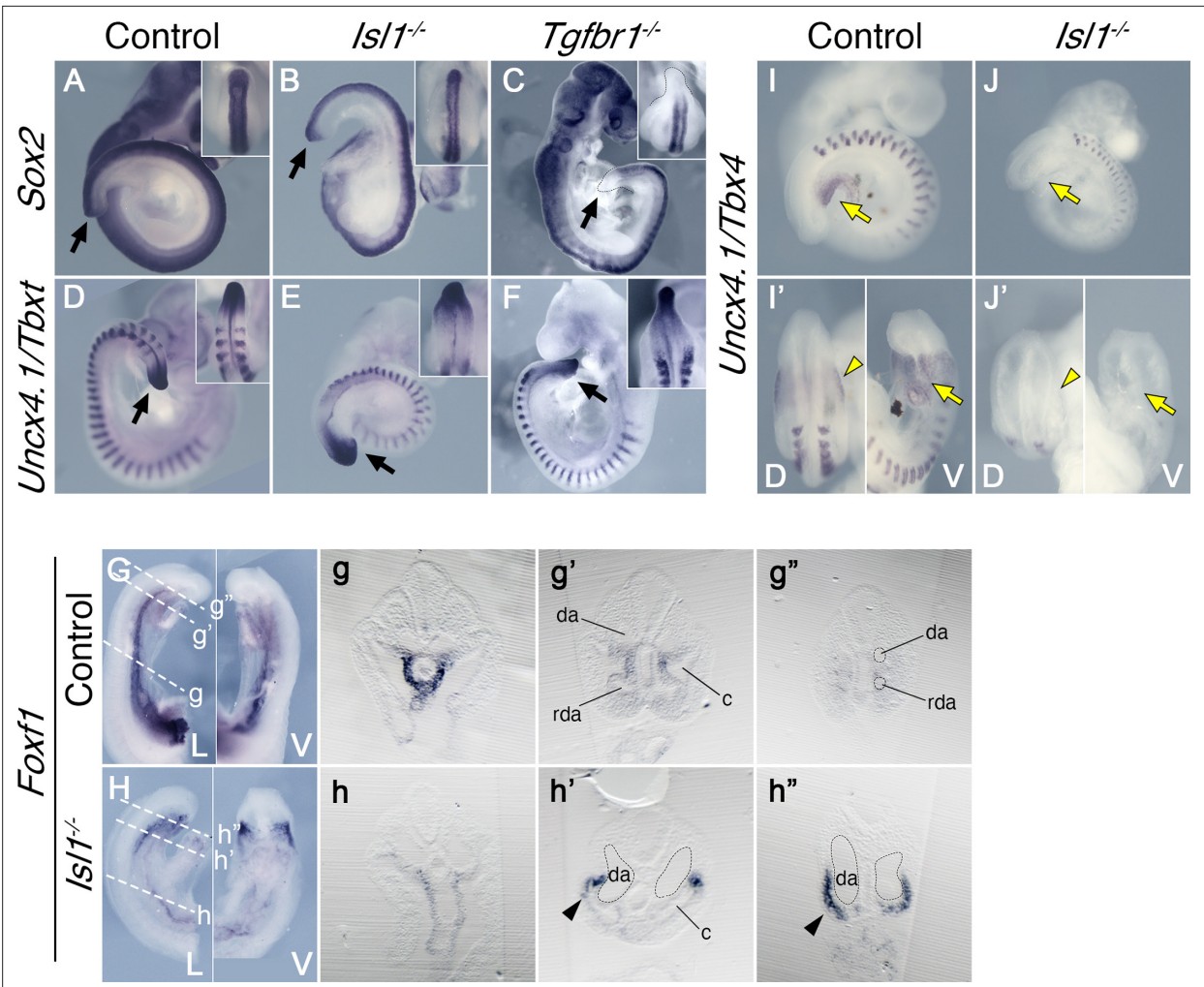

**Figure 4.** Effects of Tgfbr1 in the lateral plate mesoderm (LPM), but not in the tail bud, are mediated by Isl1. Whole-mount in situ hybridization showing expression of Sox2 (**A–C**) and Uncx4.1/Tbxt (**D–F**) in the E9.5 control (**A, D**), Isl1$^{-/-}$ (**B, E**), and Tgfbr1$^{-/-}$ (**C, F**) embryos. Isl1$^{-/-}$ embryos form tail bud (black arrows), unlike Tgfbr1$^{-/-}$ embryos. Insets in the right top corners show dorsal view of the tail bud region. (**G, h"**) Whole-mount in situ hybridization showing expression of Foxf1 in the E9.5 control (**G**) and Isl1$^{-/-}$ (**H**) embryos. g–g" and h–h" show transversal sections through the regions marked by the dashed line in G and H. Foxf1 is ectopically expressed in the splanchnopleure of the posterior region of the Isl1$^{-/-}$ (black arrowhead in h' and h"). (**I–J'**) Whole-mount in situ hybridization showing expression of Uncx4.1/Tbx4 in E9,5 control (**I, I'**) and Isl1$^{-/-}$ (**J, J'**) embryos. Tbx4 is not expressed in pericloacal mesenchyme (yellow arow) and hindlimb buds (yellow arrowheads) of Isl1$^{-/-}$ mutants. da – dorsal aorta, rda – recurved dorsal aorta, c – coelomic cavity, D – dorsal, V – ventral.

posterior embryonic regions (*Figure 4h', h"*). In addition, the DA were expanded into two globular vessels on either side of the gut tube reaching the caudal embryonic end where they merged ventrally (*Figure 4h,h', Figure 5C,c,D,d*). The connection between umbilical artery and the expanded DAs was highly disorganized (*Figure 5D*). Together, these observations indicate that *Isl1* acts downstream of *Tgfbr1* to regulate the processes associated with the trunk to tail transition in the LPM, including the main vascular system, and are consistent with the *Foxf1*-positive area of the posterior PS/allantois being a functional target of Tgfbr1 signaling to reorganize the LPM during the trunk to tail transition.

Molecular analyses showed that *Tbx4* was not expressed in the posterior part of *Isl1* mutants (*Figure 4I, I', J, J'*), indicating that *Tbx4* is downstream of *Isl1* in the regulatory network controlling the hindlimb/external genitalia, consistent with previous observations using a conditional mutant for *Isl1* (*Itou et al., 2012; Kawakami et al., 2011*).

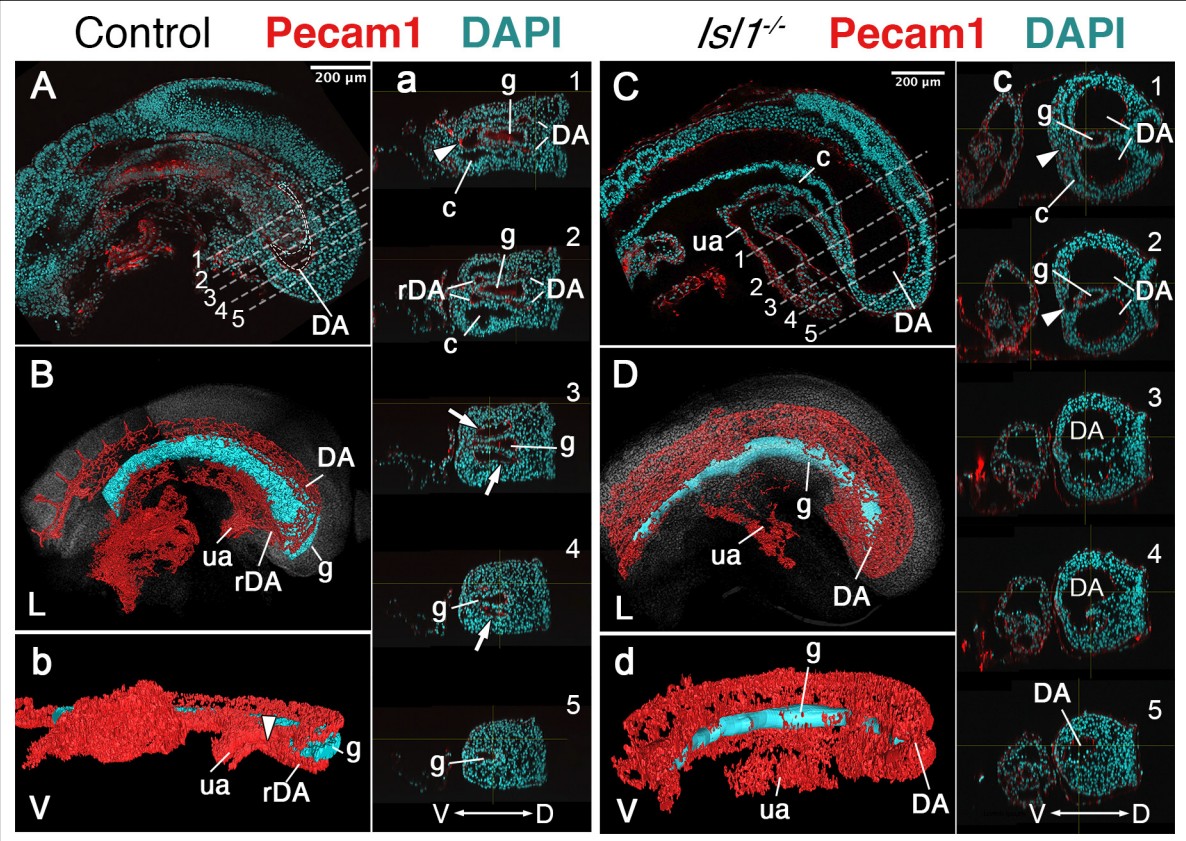

**Figure 5.** Main vascular tree of the Isl1⁻/⁻ embryos. Whole-mount immunostaining for Pecam1 (red) labeling endothelial cells in E9.5 control (**A**) and mutant (**C**) embryos. (**a, c**) Optical transversal sections through regions marked by the dashed lines in A and C. (**B, b, D, d**) 3D reconstruction of the main vascular tree (red) and the gut (cyan) of the immunostaining shown in (**A, a, C, c**). In the mutant recurved dorsal aorta (rDA) is underdeveloped and connection between dorsal aortae (DA) and the umbilical artery (ua) is established by a small vessel (white arrowhead in c1 and c2). Branches of DA are enlarged in the Isl1⁻/⁻ and merge together at the posterior end and ventral to the gut c3–5, (**d**). D – dorsal, L – lateral, V – ventral, c – coelomic cavity, g – gut.

## Proper development of the embryonic endoderm requires *Tgfbr1*

Analysis of transverse sections of the *Tgfbr1* mutant embryos suggested abnormal morphogenesis of the gut tube. Consistent with this, expression of two endodermal markers, *Foxa2* and *Shh* (**Ang et al., 1993**; **Echelard et al., 1993**) at E9.5 showed abnormal morphology at the posterior end of the gut tube (**Figure 6A, a, B, b**, **Figure 6—figure supplement 1**). This abnormal morphology was clearer in mutant embryos immunostained with the epithelial marker Keratin 8 (**Runck et al., 2014**), where it was observed that the endodermal tube finished contacting the ventral ectoderm forming a structure reminiscent of the cloacal membrane (**Figure 6C, D**). Also, in contrast with what was observed in wild-type controls, *Tgfbr1*⁻/⁻ embryos lacked the endodermal widening characteristic of the developing cloaca and failed to extend the endodermal tube caudal to the cloacal membrane to form the tail gut (**Figure 6C, D**). Remarkably, expression of the endodermal marker *Apela Hassan et al., 2010* followed abnormal patterns in *Tgfbr1* mutants. Contrary to wild-type embryos, in which the tail endodermal tube was strongly positive for *Apela* (**Figure 6E, e, F, f**), in the *Tgfbr1* mutants most of the endodermal tube was negative for this marker, its expression being observed only in a few cells in the dorsal part of the gut tube (**Figure 6E', e', F', f'**). Surprisingly, *Apela*-positive cells were found mixed with the cells of the expanded LPM (**Figure 6E', F'**), suggesting that endodermal progenitors were produced but misrouted, failing to enter the gut tube. Consistently with the endodermal origin of the *Apela*-positive cells, we observed Keratin 8 staining scattered within the extended LPM of the mutant embryos (**Figure 6c, d**).

It has been shown that the visceral endoderm contributes to the formation of the embryonic gut, with the hindgut being particularly populated by this extraembryonic tissue (**Kwon et al., 2008**). To

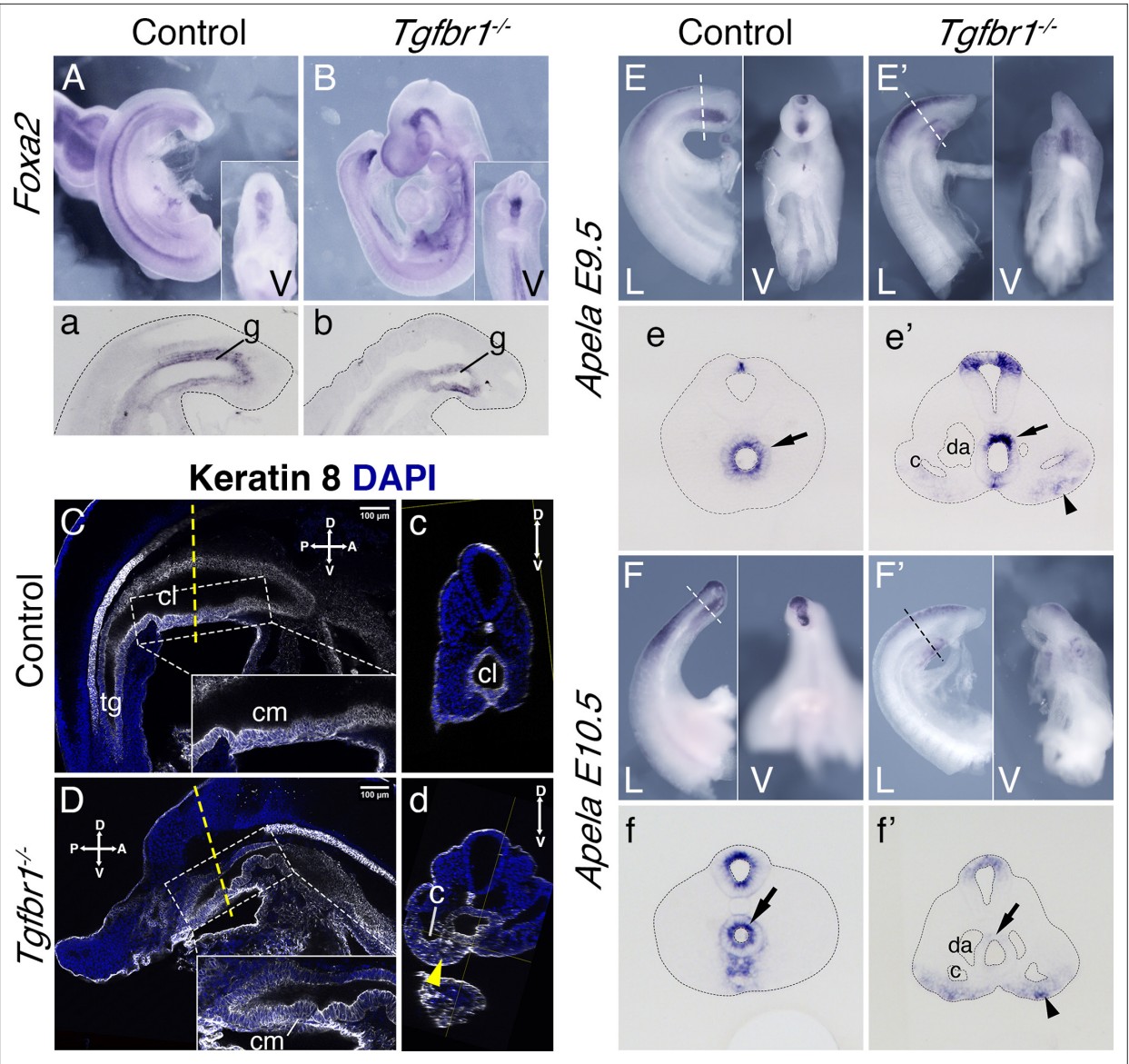

**Figure 6.** Endoderm of the Tgfbr1 KO. Expression of Foxa2 in E9.5 control (**A, a**) and Tgfbr1 KO (**B, b**) embryos. a and b show sagittal sections though the tail region. Keratin 8 staining of the cloaca region in the control (**C, c**) and Tgfbr1$^{-/-}$ (**D, d**) E10.5 embryos. Tgfbr1$^{-/-}$ do not initiate enlargement of the cloacal cavity. Insets show higher magnification of the cloacal membrane (cm). c and d show transversal optical sections marked by the dashed line in C and D. Yellow arrowhead in d shows Keratin 8 staining in expanded lateral plate mesoderm (LPM) of the Tgfbr1 mutant embryo. Apela expression in the posterior region of the E9.5 control (**E, e**) and mutant (**E', e'**) embryos. e and e' show transversal sections of regions marked by the dashed line in E and E'. Apela expression in the posterior region of the E10.5 control (**F, f**) and mutant (**F', f'**) embryos. f and f' show transversal sections of regions marked by the dashed line in F and F'. Black arrow – gut endoderm, black arrowhead – Apela-expressing cells in LPM of the mutant embryo. V – ventral, L – lateral, cl – cloaca, c – coelomic cavity, da – dorsal aorta, g – gut, hg – hindgut.

The online version of this article includes the following figure supplement(s) for figure 6:

**Figure supplement 1.** Whole-mount in situ hybridization on E9.5 wild-type (**A**) and Tgfbr1$^{-/-}$ (**B**) embryos with a probe for Shh.

understand whether the absence of cloacal and tail gut structures in the *Tgfbr1* mutants resulted from the inability of the posterior visceral endoderm cells to become incorporated into the embryonic definitive endoderm, becoming instead mixed with the mis-patterned LPM cells, we introduced the visceral endoderm reporter *Afp-GFP* transgene (**Kwon et al., 2008**) into the *Tgfbr1* mutant background. Analysis of E7.5 *Afp-GFP$^{+/0}$:Tgfbr1$^{-/-}$* embryos indicated that the dispersal of visceral endodermal cells was not affected by the absence of *Tgfbr1* (**Figure 7—figure supplement 1**). In addition, Afp-GFP-positive visceral endoderm cells were observed in the embryonic endoderm of

*Tgfbr1* mutant embryos at E8.5, in a distribution comparable to wild-type embryos (*Figure 7A, A', B, B'*). At later stages of development, however, when embryos engage in tail development, distinct distributions were observed in the wild-type and mutant embryos. Both at E9.5 and E10.5, the entire endodermal tube of *Tgfbr1*$^{-/-}$ embryos contained GFP-positive cells, also showing the premature end at the ventral surface of the embryo and the absence of further posterior extension to form the tail gut (*Figure 7D–D", F, F'*). Importantly, we did not observe Afp-GFP signal mixed with the extended LPM (*Figure 7F", F'''*). These data thus indicate that the absence of *Tgfbr1* does not affect recruitment of visceral endodermal cells to the embryonic gut, and that the abnormal *Apela* patterns observed in *Tgfbr1* mutant embryos are unlikely to derive from misrouting of visceral endodermal cells.

Interestingly, while we observed a significant contribution of Afp-GFP-positive visceral endoderm cells to the embryonic endoderm of wild-type embryos at E8.5 (*Figure 7A, A'*), we could detect just a few cells in the tail gut of E9.5 embryos (*Figure 7C–C''*) and virtually none at E10.5 (*Figure 7E, E'*). This indicates that, while the visceral endoderm contributes significantly to the embryonic gut up to the region of the cloaca, as previously reported (*Kwon et al., 2008*), it is likely to play a minor role in the extension of the endodermal tube growing into the tail. The *Apela* expression patterns indicate that this gene is active in the newly generated endodermal tissues, becoming progressively downregulated after they are part of the gut tube (*Hassan et al., 2010*). The strong *Apela* expression restricted to the posterior portion of the endodermal tube within the developing tail (*Figure 8A, B, Bb*), suggests that the tail gut grows from the addition of cells at the tip of this structure. Interestingly, analysis of the *T-str-creERT:ROSA26-R-gal* reporter activity upon tamoxifen administration at E8.0 showed the presence of β-galactosidase-positive cells in the tail gut (*Figure 3d*). A similar finding has also been independently reported using a different T-creERT strain (*Anderson et al., 2013*). Of note, more anterior regions of the gut were negative for β-galactosidase under these conditions, despite the presence of labeled cells in adjacent mesodermal tissues (*Figure 3c*). This suggests that the most caudal region of the gut tube grows through the addition of cells generated from a structure derived from the PS located at the end of the growing tail. Consistent with this hypothesis, we observed an *Apela*-positive structure adjacent to the tip of the gut at the posterior end of the growing tail (*Figure 8—figure supplement 1C*). A similar structure was observed in embryos immunostained for Keratin 8 (*Figure 8*; *Figure supplement 1A–b*).

We further tested whether this *Apela*- and Keratin 8-positive region at the tip of the tail bud contributes to the gut tube using a DiI-mediated cell tracing approach ex vivo. E9.5 embryos injected with DiI that showed no label in the gut tube just after injection (*Figure 8—figure supplement 2A-c3*) were analyzed after 20 hr in culture (*Figure 8E–h'*, *Figure 8—figure supplement 2*). In the three embryos that filled this criterium, DiI-positive cells were observed in the gut tube epithelium. Additionally, staining was observed in the surrounding mesenchyme and in the neural tube, possibly due to DiI labeling also becoming incorporated into the NMC population. Even considering the possible leakage of the DiI into the NMC compartment, the presence of DiI signal in the gut epithelium further supports the existence of a region at the tip of the tail bud able to generate cells that become incorporated into the tail gut.

## Discussion

The transition from trunk to tail development involves major tissue reorganization affecting all germ layers. Formation of spinal cord and somitic mesoderm is maintained by the relocation of the neural-mesodermal-competent cells from the caudal lateral epiblast CLE in the epiblast into the CNH in the tail bud through an incomplete EMT triggered by the concurrent activity of Tgfbr1 and Snai1 (*Cambray and Wilson, 2007*; *Dias et al., 2020*; *Henrique et al., 2015*; *Wilson et al., 2009*; *Wymeersch et al., 2021*; *Wymeersch et al., 2019*). The progenitors of the lateral mesoderm, however, undergo a process of terminal differentiation resulting in the formation of the primordia of the hindlimb and of the external genitalia (*Jurberg et al., 2013*). Genetic analyses indicate that *Tgfbr1* signaling is both necessary and sufficient to activate the mechanisms regulating those terminal differentiation processes, as illustrated by their premature activation in transgenic gain of function experiments (*Jurberg et al., 2013*) and the absence of early markers for the primordia of the hindlimb or the GT in *Tgfbr1* null mutant embryos (*Dias et al., 2020*). Our data now show that the requirement of *Tgfbr1* encompasses the development of many other tissues undergoing a morphological and functional

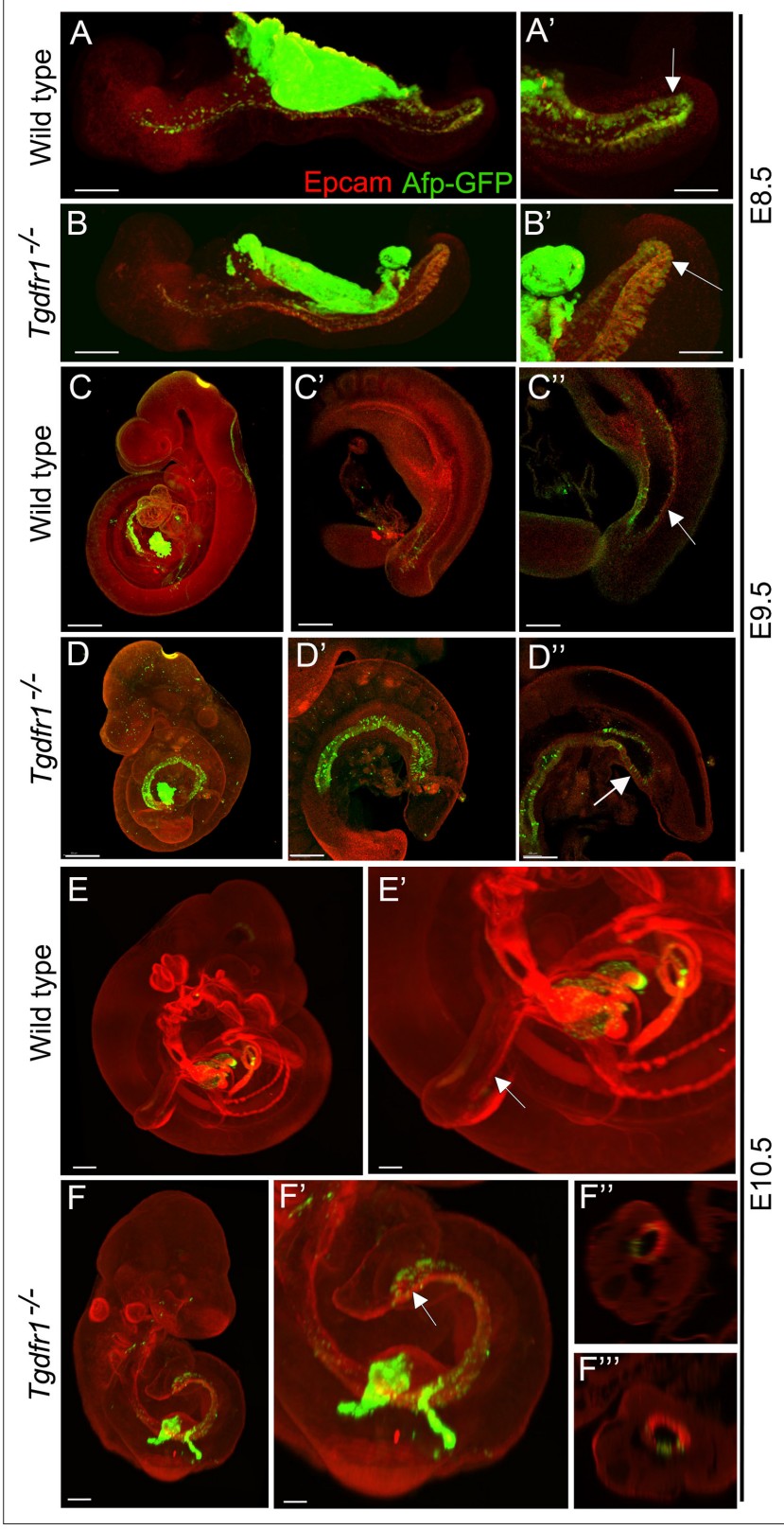

**Figure 7.** Analysis of the contribution of the visceral endoderm to the embryonic gut. GFP expression from the Afp-GFP transgenics was analyzed at E8.5 (**A–B'**), E9.5 (**C–D"**), or E10.5 (**E–F'"**) in wild-type (**A, A', C–C", E, E'**) or *Tgfbr1⁻/⁻* (**B, B', D–D", F–F'"**) embryos. (**C'** and **D'**) show a 3D image of the embryo, and **C"** and **D"** show transversal sections. (**F'** and **F"**) show transversal sections through the caudal part of F'. The embryonic endoderm

*Figure 7 continued on next page*

*Figure 7 continued*

was labeled by immunofluorescence against Epcam. Arrows in **A'**, **B'** indicate the hindgut; arrows in **C"** and **E'** indicate the tail gut; arrows in **D"** and **F'** indicate the cloacal membrane. Size bars: A, B: 200 μm; A', B': 100 μm; C, D: 300 μm; C', D': 200 μm; C", D": 150 μm; E: 300 μm; F: 200 μm; E': 150 μm; F': 100 μm.

The online version of this article includes the following figure supplement(s) for figure 7:

**Figure supplement 1.** Analysis of visceral endoderm (VE) dispersal in wild-type and *Tgfbr1* mutant E7.5 embryos.

reorganization during the trunk to tail transition, including the major vascular system and the embryonic endoderm. These observations place *Tgfbr1* as a master regulator of the trunk to tail transition.

An interesting conclusion from our work is the identification of the posterior PS/allantois as a candidate for the structure mediating the different *Tgfbr1*-dependent processes involving the lateral mesoderm and endoderm during the trunk to tail transition. Cell tracing experiments identified the posterior epiblast/PS as the region providing the cells building the trunk LPM (***Wymeersch et al., 2016***), being the posterior PS abutting the allantois also involved in organizing the recruitment of visceral endodermal cells to the embryonic gut tube (***Rodriguez and Downs, 2017***). The transition to tail development entails the fading of the epiblast and PS, as they become replaced by the tail bud as the main driver of axial extension. At this stage, the allantois also leaves its position at the posterior end of the embryo to occupy more anterior and ventral positions while organizing the connection between embryonic and extraembryonic structures (***Arora and Papaioannou, 2012***; ***Downs and Rodriguez, 2020***). Gene expression analysis shows that genes involved in vascular morphogenesis are specifically expressed in the posterior epiblast/PS region (***Wymeersch et al., 2019***). Cell tracing experiments indicate that the posterior epiblast/PS, which lays down the LPM during trunk formation, generates the VLM posterior to the allantois during the trunk to tail transition (***Wymeersch et al., 2016***) and that these cells contribute to the primordium of the external genitalia later in development (***Tschopp et al., 2014***). The involvement of the PS in the formation of the VLM and genital primordia is also supported by our reporter data with the *T-str-creERT* transgenic line and *Isl1* cell lineage analyses (***Yang et al., 2006***). The absence of VLM in the *Tgfbr1* mutants indicates that signaling through this receptor is required to organize the proper switch of the posterior epiblast/PS from a trunk developmental mode, involving entering VLM fates. *Foxf1*, which is expressed in the posterior PS, maintains expression in the derivatives of this structure after the transition to tail development (***Figure 2—figure supplement 1***; ***Astorga and Carlsson, 2007***). We suggest that the strong and expanded *Foxf1* expression throughout the posterior end of the extended LPM of the *Tgfbr1* mutant embryos represents a molecular vestige of posterior PS that failed to form the VLM, and instead became trapped within mis-patterned LPM extending from the trunk. As the posterior PS is also thought to play a relevant role in the connection of the paired DAs with the allantois artery to link the embryonic and extraembryonic vascular systems (***Downs and Rodriguez, 2020***), defective posterior PS reorganization during the trunk to tail transition could explain the DA abnormalities observed in the *Tgfbr1* mutant embryos.

The embryonic tail gut has received little attention and mechanisms of its growth remain largely unknown. Analysis of *Tgfbr1⁻/⁻* embryos revealed that signaling through this receptor is essential for the development of the endodermal tube posterior to the cloacal plate and potentially the cloaca itself. *Apela* expression in wild-type embryos is high in the newly formed endoderm, becoming downregulated as the endodermal tube differentiates (***Hassan et al., 2010***). The presence of the highest *Apela* levels in the posterior part of the tail gut at different embryonic stages (***Hassan et al., 2010***) suggests that this structure grows through the addition of new tissue at its posterior end. The low levels of *Apela* expression in the endoderm of the *Tgfbr1* mutants might thus indicate that this structure was generated earlier in development, during the phase of trunk extension. In addition, the presence of *Apela* signal mixed with the LPM suggests that new endodermal cells are still produced but failed to enter the gut, being instead mistargeted to the mesoderm. The absence of fluorescence signal in the mesodermal tissue of *Tgfbr1⁻/⁻:Afp-GFP* embryos indicates that those cells are most likely not derived from the posterior visceral endoderm (***Kwon et al., 2008***). Interestingly, the similarities between the *Apela* and *Foxf1* expression in the posterior region of *Tgfbr1* mutant embryos suggest that their developmental history might be somehow linked. Given the role of the posterior PS abutting the allantois in the recruitment of visceral endodermal cells to the embryonic gut (***Rodriguez and Downs, 2017***), it is possible that tail gut growth is organized by a structure derived from a specific

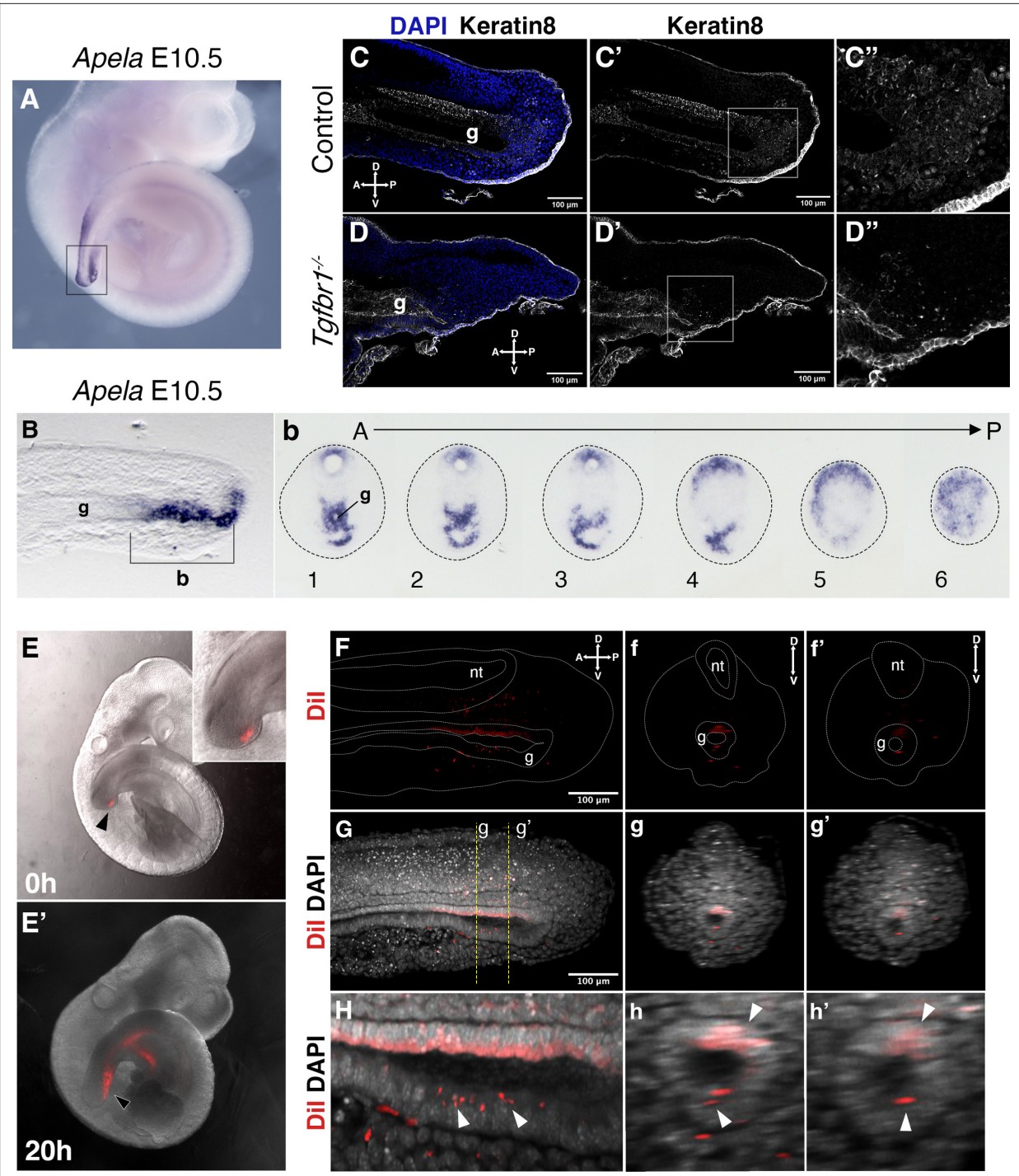

**Figure 8.** Tail gut endoderm has contribution from the posterior pool of tail bud cells. (**A**) Whole-mount in situ hybridization showing expression of *Apela* in E10.5 wild-type embryo. (**B**) Sagittal section through the region marked by rectangle in A shows presence of *Apela*-stained structure posterior to tail gut endoderm. (**b**) Series of transversal sections though the *Apela*-expressing region posterior to the gut endoderm (marked by square bracket in **B**). (**C–D"**) Sagittal optical sections through the tail region of the whole-mount immunostaining for Keratin 8 (white) in E10.5 control (**C–C"**) and *Tgfbr1*⁻/⁻ (**D–D"**) embryos. Nuclei are shown in blue. Squares in **C'** and **D'** show the pool of epithelial cells posterior to the tail gut tube. This region coincides with newly formed endodermal cells expressing *Apela* shown in **B, b**. **C".** D" higher magnification of the region marked by square in **C'** and **D'**. A – anterior, P – posterior, g – gut. (**E, E'**) Whole-mount images of embryos injected with DiI in the Apela-positive region of the tail bud posterior to the gut endoderm, just after injection (**E**) or after 20 hr of incubation (**E'**). The magnification of the tail bud in the inset shows the absence of label in the gut tube. (**F–h'**) Optical sections from two-photon images of the embryo in **E'** to show the presence of DiI cells in the gut tube. (**F–H**) show sagittal sections;

*Figure 8 continued on next page*

*Figure 8 continued*

f–h' show transverse sections. (**F–f"**) show the DiI channel; tail, neural tube (nt) and gut (g) are outlined with a dashed line. (**G–g'**) show DiI and DAPI channels together. **H–h'** show magnification of DiI labeling in the gut tube (white arrowheads).

The online version of this article includes the following figure supplement(s) for figure 8:

**Figure supplement 1.** Tail gut of E9.5 wild-type embryo.

**Figure supplement 2.** DiI labeling of the *Apela*-positive region in the E9.5 tail bud.

region of the posterior PS that enters the tail during the transition to tail development. The existence of a region at the tip of the tail bud that feeds cells to the gut tube is supported by our DiI tracing experiment. Understanding the cellular identity and developmental potential of this region requires further investigation. In addition, the observation that the ROSA26 reporter labels the tail gut when activated by the *T-str-creERT* driver at the stage of the trunk to tail transition (*Anderson et al., 2013*; *Figure 3d*) is consistent with the involvement of the PS or a derivative of this structure in the formation of the gut tube posterior to the cloacal membrane. Were this the case, the transition of this PS-derived structure should be under *Tgfbr1* control.

Taken together, this work along with previous studies, suggests that the control of the trunk to tail transition by *Tgfbr1* entails two distinct, and apparently independent, components acting on two areas of the epiblast/PS. The first component involves a cooperation between *Tgfbr1* and *Snai1* to organize the relocation of NMC progenitors from the CLE to the CNH through a partial EMT (*Dias et al., 2020*). This component is also associated with the generation of the tail bud that replaces the epiblast/PS as the driver of axial elongation (*Wilson et al., 2009*; *Wymeersch et al., 2021*). The second component would target the posterior part of the epiblast/PS containing the progenitors for the lateral mesoderm as well as an organizing center for endodermal development (summarized in

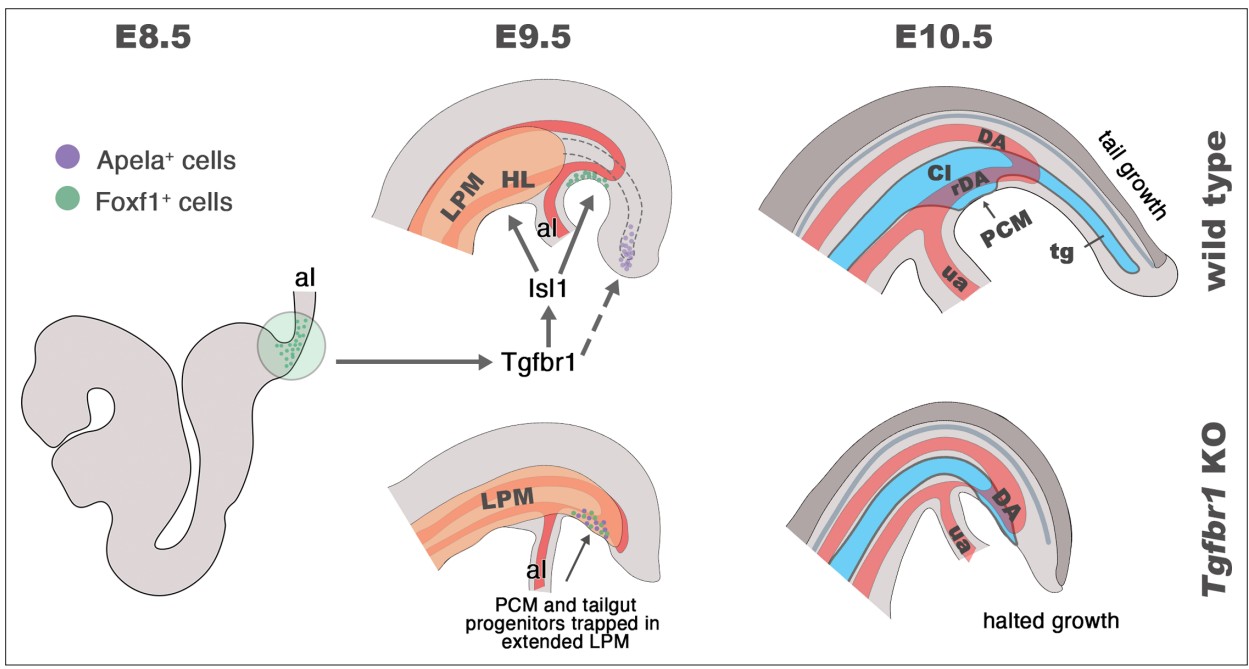

**Figure 9.** Schematic representation of *Tgfbr1* activity on the posterior epiblast/primitive streak (PS) region during the trunk to tail transition. At E8.5 embryo undergoes turning, associated with anterior relocation of the allantois along the ventral side of the embryo. In wild-type embryos (top panel), *Tgfbr1* acts upstream of *Isl1*, which induces hindlimb (HL) formation from the somatic lateral plate mesoderm (LPM), marking the posterior limit of this mesodermal compartment. Additionally, *Isl1* is involved in formation of the pericloacal mesenchyme (PCM), likely from the *Foxf1+* in the posterior PS (shown in green), and in the development of the recurved dorsal aorta (rDA). Tail gut (tg) growth is, at least in part, supplied by the cells located in the tail bud (endodermal *Apela+* cells are shown in purple). At E10.5 the trunk to tail transition is completed, resulting in the formation of the posterior trunk structures, including HL, cloaca (Cl), PCM, and the connection of embryonic/extraembryonic (umbilical artery – ua) blood circulation via rDA. Tail growth continues generating neural tube (dark gray), presomitic mesoderm (not shown), and tg (blue). In the absence of *Tgfbr1* (bottom panel) *Isl1* is not activated, hindlimbs are not induced from the LPM, which, instead, keeps extending posteriorly. PCM and tg progenitor cells are misrouted and trapped in the posteriorly extended LPM. The rDA is underdeveloped. Development of *Tgfbr1* mutants is halted around E10.5.

**Table 1.** Genotyping primers.

**Genotyping primers**

| | | |
|---|---|---|
| *Tgfbr1* mutant allele | Forward | CTACTGTGTTTCAAATGGGAGGGC |
| | Reverse | GGCCTGTCGGATCCTATCATC |
| *Tgfbr1* wild-type allele | Forward | CTACTGTGTTTCAAATGGGAGGGC |
| | Reverse | ACATACAAATGGCCTGTCTCG |
| *Isl1* mutant allele | Forward | GCCACTATTTGCCACCTAGC |
| | Reverse | AGGCAAATTTTGGTGTACGG |
| *Isl1* wild-type allele | Forward | GCCACTATTTGCCACCTAGC |
| | Reverse | CAAATCCAAAGAGCCCTGTC |
| Cre recombinase | Forward | CGAGTGATGAGGTTCGCAAG |
| | Reverse | CCTGATCCTGGCAATTTCGGCT |

*Figure 9*). Here, *Tgfbr1* activity triggers a combination of programs leading to the organization of the exit channels of the intestinal and urogenital systems, as well as the connection of the embryonic and extraembryonic circulation, and the formation of the hindlimbs and external genitalia. The finding that *Isl1* mutants exhibit many of the features observed in the lateral mesoderm and vascular system of *Tgfbr1* mutants, together with the absence of *Isl1* expression in *Tgfbr1* mutants (*Dias et al., 2020*) identifies *Isl1* as a key downstream mediator of the second component of *Tgfbr1* activity controlling the trunk to tail transition. Additional work will be required to elucidate the mechanisms regulating the fate and cell dynamics of the posterior epiblast/PS during the trunk to tail transition.

## Materials and methods
### Mouse lines and embryos

The *Tgfbr1*[+/−] (*Dias et al., 2020*), *Alf-GFP* (*Kwon et al., 2008*), *ROSA26-R-gal* (Jackson Labs stock #003474, RRID:IMSR_JAX:003474; *Soriano, 1999*), *ROSA26-R-EYFP* (Jackson Labs stock #006148, RRID:IMSR_JAX:006148; *Srinivas et al., 2001*), and *Isl1-cre* Jackson Labs stock #024242, RRID:IMSR_JAX:024242; *Yang et al., 2006* used in this work have been previously described. *T-str-creERT* was generated by cloning the *creERT* cDNA, containing the SV40 polyadenylation signal obtained from the *Cdx2-creERT* construct (*Jurberg et al., 2013*), under the control of the PS enhancer of the Brachyury (*Tbxt*) gene (*Clements et al., 1996*). The construct was used to generate transgenic animals by pronuclear microinjection according to standard protocols (*Hogan et al., 1994*). Mutant and transgenic lines were genotyped from ear or digit biopsies incubated in 50 µl of PBND buffer (50 mM KCl, 10 mM Tris–HCl, pH 8.3, 2.5 mM MgCl₂, 0.1 mg/ml gelatin, 0.45% NP40, 0.45% Tween-20) supplemented with 100 µg/ml of proteinase K at 55°C overnight. Samples were incubated at 95°C for 15 min to heat-deactivate proteinase K. 1 µl of genomic DNA was used in PCR reaction with the relevant primers specified in *Table 1*.

*Tgfbr1* null and *Isl1* null embryos were generated from intercrosses between *Tgfbr1*[+/−] and *Isl1*[+/cre] mice, respectively. Embryos obtained from heterozygous crossings were genotyped from their yolk sacs. Yolk sacs were collected to 50 µl of lysis buffer (50 mM KCl, 10 mM Tris–HCl, pH8.3, 2 mM MgCl₂, 0.45% Tween-20, 0.45% NP40) supplemented with 100 µg/ml of proteinase K and incubated at 55°C overnight. Samples were heat-deactivated as described above. PCR was performed using 1 µl of genomic DNA using the primers specified in *Table 1*.

All animal procedures were performed in accordance with Portuguese (Portaria 1005/92) and European (directive 2010/63/EU) legislations and guidance on animal use in bioscience research. The project was reviewed and approved by the Ethics Committee of 'Instituto Gulbenkian de Ciência' and by the Portuguese National Entity 'Direcção Geral de Alimentação Veterinária' (license reference: 014308).

## Whole-mount in situ hybridization and sectioning

Embryos were fixed in 4% paraformaldehyde in PBS (PFA) overnight, then dehydrated through a 25%, 50%, and 75% series of methanol in PBT (PBS, 0.1% Tween-20), then incubated in 100% methanol. Embryos were then rehydrated through a reverse methanol/PBT series and incubated three times in PBT for at least 5 min each at room temperature. Embryos were then bleached for 1 hr in 6% hydrogen peroxide in PBT and permeabilized in 10 µg/ml of proteinase K (Roche #3115801001) in PBT for a time period that depended on the embryo size. The reaction was then quenched with a 2-mg/ml solution of glycine in PBT, washed twice in PBT and postfixed in a 4% PFA and 0,2% glutaraldehyde mix for 20 min, followed by two washes in PBT. Hybridization was performed at 65°C overnight in hybridization solution (50% formamide, 1.3× saline sodium citrate (SSC) pH 5.5 [20× SSC is 3 M NaCl, 300 mM sodium citrate], 5 mM EDTA, 0.2% Tween-20, 50 µg/ml yeast tRNA, 100 µg/ml heparin) containing the relevant digoxigenin-labeled antisense RNA probes. RNA probes were in vitro transcribed from the linearized vector for 3 hr at 37°C with the corresponding RNA polymerase and DIG RNA Labeling Mix (Roche #11277073910). The reaction product was verified in 0.8% agarose gel and diluted in hybridization solution for further use. After hybridization, embryos were washed twice at 65°C with hybridization solution without tRNA, heparin, and the RNA probe and then in a 1:1 mix of hybridization solution and TBST (25 mM Tris–HCl, pH 8.0, 140 mM NaCl, 2.7 mM KCl, 0.1% Tween-20) for 30 min at 65°C. Embryos were then washed three times with TBST at room temperature, equilibrated in MABT (100 mM maleic acid, 150 mM NaCl, 0.1% Tween-20, pH 7.5) and blocked in MABT blocking buffer [MABT containing 1% blocking reagent (Roche #11096176001)] with 10% sheep serum for 2.5 hr at room temperature. Embryos were then incubated overnight at 4°C with a 1:2000 dilution of alkaline phosphatase-conjugated anti-digoxigenin antibody (Roche #11093274910, RRID:AB_514497) in MABT blocking buffer with 1% sheep serum. After extensive washes with MABT at room temperature, embryos were equilibrated in NTMT buffer (100 mM Tris–HCl, pH 9.5, 50 mM MgCl$_2$, 100 mM NaCl, 0.1% Tween-20) and developed with a 1:50 dilution of NBT/BCIP solution (Roche #11681451001) in NTMT at room temperature in the dark. Stained embryos were mounted in a 0.45% gelatin, 27% bovine serum albumin (BSA), 18% sucrose mix, jellified with 1.75% glutaraldehyde and sectioned at 35 µm on a Leica Vibratome VT 1000 S. At least three embryos of each relevant genotype were analyzed with each probe.

## Whole-mount immunofluorescence and image processing

Embryos were fixed in 4% PFA on ice for 2 hr and then dehydrated through a 25%, 50%, and 75% methanol/PBST (PBS, 0.1% Triton X-100) series followed by 100% methanol. Embryos were then rehydrated through a reverse methanol PBST series, washed with PBST and permeabilized in 0.5% Triton X-100 in PBS for 1 hr and incubated in 1 M glycine in PBST for 30 min to reduce unspecific binding. After several washes in PBST embryos were blocked in 1% BSA, 3% donkey serum in PBST at 4°C overnight. Embryos were then incubated for 72 hr at 4°C with the following dilutions of the primary antibodies in blocking buffer: Pecam1 1:50 (Abcam #ab28364, RRID:AB_726362), Keratin 8 1:100 (Troma1, developed by Dr Brulet and Dr Kemler, obtained from the NICHD Developmental Studies Hybridoma Bank maintained by the University of Iowa, RRID:AB_2891089). Secondary antibodies were diluted 1:1000 in blocking buffer and embryos incubated for 48 hr at 4°C. After extensive washes in PBST embryos were stained with a 1:10,000 DAPI dilution in PSBT at 4°C overnight. Embryos were then mounted on a depression slide with RapiClear 1.49 clearing reagent (SunJin lab). Embryos were imaged on a Prairie Multiphoton microscope using an Olympus 20× 1.0 NA W objective. Stacks were then digitally stitched in Fiji using the Grid/Collection stitching plugin. After removing the outliers, tissues were segmented using Amira Software. Three biological replicates were performed per genotype.

## Analysis of the contribution of the visceral endoderm to the embryonic gut

E7.5 embryos were fixed for 20 min in 4% PFA at room temperature, washed three times in PBST, and then counterstained in 5 µg/ml Hoechst and 5 U/ml phalloidin before imaging. E8.5 embryos were permeabilized in 0.5% Triton X-100 in PBS for 20 min, washed three times in PBST and incubated in blocking buffer containing 5% donkey serum (Jackson Labs) and 1% BSA in PBST for 1 hr at 4°C. Embryos were then incubated overnight at 4°C with Epcam (Biolegend, #118202, RRID:AB_1089027) (1:100) and GFP (Aveslabs #GFP-1020) (1:500) in blocking buffer. After three washes in PBST, embryos

were incubated with secondary antibody (1:500) overnight at 4°C, and then washed again three times in PBST and counterstained in 5 µg/ml Hoechst. For E9.5 and E10.5 embryos, samples were fixed for 1 hr in 4% PFA at room temperature, washed three times in PBS, then dehydrated through a 25%, 50%, and 75% methanol/$H_2O$ series followed by 100% methanol. Embryos were stained with antibodies against Epcam (1:100) and GFP (1:500) using the iDISCO+ tissue clearing protocol as previously described (*Renier et al., 2014*) (updated at https://idisco.info/). E7.5 embryos were mounted in PBS and imaged on a Zeiss LSM880 using a Plan-Apo 20×/NA0.8 M27 objective. E8.5 embryos were mounted in FocusClear clearing reagent and imaged on a Zeiss LSM 880 using a Plan-Apo 20×/NA0.8 M27 objective. E9.5 embryos were mounted in dibenzyl ether and imaged on a Zeiss LSM 880 using a Plan-Apo 20×/NA0.8 M27 objective and an EC Plan-NEOFLUAR 10×/NA0.3 objective. E10.5 embryos were imaged on a Luxendo MuVi SPIM light-sheet microscope using a Nikon Plan-Apo 10×/NA0.8 Glyc objective. Raw image data were processed in ZEN (Zeiss), Luxendo Image Processor, or Imaris (Bitplane, http://www.bitplane.com/) software.

## Ex vivo culture with DiI labeling

E9.5 embryos were dissected out in ice cold media [DMEM (Gibco, Life Technologies #11965092)/15% fetal bovine serum (FBS; Gibco, Life Technologies, #A5670701)/11 mM HEPES (Sigma #7365-45-9)]. Stock DiI solution was prepared by dissolving aliquot of powdered CellTracker CM-DiI (Life technologies, #C7000) in 10 µl of 96% ethanol. Working labeling solution was a 1:10 dilution of the stock solution in 0.3 M sucrose (Sigma). Embryos were injected with labeling solution in the ventral tail bud by mouth pipetting using a glass capillary and collected individually in a 24-well plate on ice. Injected embryos were first imaged on SteREO Lumar.V12, Zeiss and then cultured for 20 hr at 37°C in a rotator bottle culture apparatus (B.T.C. Engineering, Milton, Cambridge, UK) at 37°C, in a 65% $O_2$ atmosphere. Each embryo was cultured individually in a tube with 1.5 ml of DMEM/F-12, GlutaMAX (Gibco, Life Technologies, #31331-028) containing 15% FBS, 11 mM HEPES (Sigma #7365-45-9) and supplemented with a penicillin/ streptomycin mixture. Cultured embryos were washed with PBS, imaged on SteREO Lumar.V12, Zeiss and fixed in 4% PFA for 1 hr at room temperature. Next, embryos were permeabilized with 0,3% Tween in PBS for 1 hr at room temperature. Nuclei were stained with a 1:5000 solution of DAPI in PBS for 16 hr at 4°C with rotation. Embryos were then washed in PBS, mounted on depression slides, and cleared with RapiClear 1.49 clearing reagent (SunJin lab). Imaging was done on a Prairie Multiphoton microscope using an Olympus 20× 1.0 NA W objective.

## Reporter tracing with T-str-creERT transgenics

Pregnant females from *T-str-creERT* and either *ROSA26-R-βgal* or *ROSA26-R-YFP* intercrosses were treated with tamoxifen (200 µl of a 1 mg/ml solution in corn oil) by oral gavage at different times from E7.5 to E8.5 and harvested at E9.5 or E10.5. When the *ROSA26-R-YFP* reporter was used, embryos were dissected on PBS and observed with SteREO Lumar.V12, Zeiss. When the *ROSA26-R-βgal* reporter was used embryos were fixed with 4% PFA at 4°C for 30 min, then washed three times in PBS containing 0.02% Tween-20 for 10 min each at room temperature and developed with 0.4 mg/ml X-gal in 5 mM $K_3Fe(CN)_6$, 5 mM $K_4Fe(CN)_6$·$3H_2O$, 2 mM $MgCl_2$, 0.02% NP40, 0.02% Tween-20, in PBS for several hours at 37°C in the dark. The reaction was stopped with wash buffer, fixed with 4% PFA overnight and sectioned as described for the whole-mount in situ-stained embryos. To test the time required to observe reporter activity upon tamoxifen treatment, pregnant females from *T-str-creERT* and *ROSA26-R-YFP* intercrosses were treated with a single tamoxifen dose and embryos recovered after 6, 8, or 10 hr after treatment and evaluated for YFP signal.

## Acknowledgements

We would like to thank the members of the Mallo lab for continuous support at different stages of this project, the IGC mouse facility for their help with animal housing, and the Mouse Genetics Core Facility of Memorial Sloan Kettering Cancer Center for rederivation of the *Tgfbr1* mouse line. This project was funded by Fundação para a Ciência e a Tecnologia (FCT) grants 2022.01629.PTDC to MM (DOI: 10.54499/2022.01629.PTDC), the PhD fellowship PD/BD/128437/2017 to AL, and the research infrastructure Congento LISBOA-01-0145-FEDER-022170 to the animal facility, co-financed by Lisboa 2020/FEDER and FCT (Portugal). AKH and YYK are supported by the NIH (R01DK127821, R01HD094868, R01HD035455, and P30CA008748).

## Additional information

### Funding

| Funder | Grant reference number | Author |
|---|---|---|
| Fundacao para a Ciencia e a Tecnologia | PD/BD/128437/2017 | Anastasiia Lozovska |
| Fundacao para a Ciencia e a Technologia | DOI: 10.54499/2022.01629.PTDC | Moises Mallo |
| National Institutes of Health | R01DK127821 | Anna-Katerina Hadjantonakis |
| National Institutes of Health | R01HD094868 | Anna-Katerina Hadjantonakis |
| National Institutes of Health | R01HD035455 | Anna-Katerina Hadjantonakis |
| National Institutes of Health | P30CA008748 | Anna-Katerina Hadjantonakis |

The funders had no role in study design, data collection and interpretation, or the decision to submit the work for publication.

### Author contributions

Anastasiia Lozovska, Conceptualization, Data curation, Investigation, Writing – original draft, Writing – review and editing; Ana Casaca, Ana Novoa, Ying-Yi Kuo, Arnon D Jurberg, Investigation; Gabriel G Martins, Visualization, Methodology; Anna-Katerina Hadjantonakis, Formal analysis, Investigation, Writing – review and editing; Moises Mallo, Conceptualization, Supervision, Funding acquisition, Writing – original draft, Writing – review and editing

### Author ORCIDs

Anastasiia Lozovska http://orcid.org/0000-0002-9842-6450
Ying-Yi Kuo https://orcid.org/0000-0002-1962-5559
Moises Mallo https://orcid.org/0000-0002-9744-0912

### Ethics

All animal procedures were performed in accordance with Portuguese (Portaria 1005/92) and European (directive 2010/63/EU) legislations and guidance on animal use in bioscience research. The project was reviewed and approved by the Ethics Committee of 'Instituto Gulbenkian de Ciência' and by the Portuguese National Entity 'Direcção Geral de Alimentação Veterinária' (license reference: 014308).

Joint Public Review: https://doi.org/10.7554/eLife.94290.3.sa1
Author response https://doi.org/10.7554/eLife.94290.3.sa2

## Additional files

### Supplementary files

MDAR checklist

### Data availability

Raw images for the figures of this manuscript have been deposited in BioImage Archive (https://www.ebi.ac.uk/bioimage-archive), Accession number S-BIAD1571, DOI:https://doi.org/10.6019/S-BIAD1571.

The following dataset was generated:

| Author(s) | Year | Dataset title | Dataset URL | Database and Identifier |
|---|---|---|---|---|
| Mallo M | 2025 | Images from the manuscript "Tgfbr1 regulates lateral plate mesoderm and endoderm reorganization during the trunk to tail transition" | https://www.ebi.ac.uk/biostudies/bioimages/studies/S-BIAD1571 | BioImage Archive, 10.6019/S-BIAD1571 |

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

## Appendix 1

**Appendix 1—key resources table**

| Reagent type (species) or resource | Designation | Source or reference | Identifiers | Additional information |
|---|---|---|---|---|
| Strain, strain background (*M. musculus*) | Isl1Cre | Jackson Labs | Stock # 024242 RRID:IMSR_JAX:024242 | *Yang et al., 2006* |
| Strain, strain background (*M. musculus*) | ROSA26-R-gal | Jackson Labs | Stock #003474, RRID:IMSR_JAX:003474 | *Soriano, 1999* |
| Strain, strain background (*M. musculus*) | ROSA26-R-EYFP | Jackson Labs | Stock #006148, RRID:IMSR_JAX:006148 | *Srinivas et al., 2001* |
| Strain, strain background (*M. musculus*) | Tgfbr1+/− | *Kwon et al., 2008* eLife 9, e56615 | | |
| Strain, strain background (*M. musculus*) | Alf-GFP | *Kwon et al., 2008* Dev. Cell 15, 509–520 | | |
| Antibody | Pecam1 | Abcam | Cat #ab28364, RRID:AB_726362 | |
| Antibody | Keratin 8 | Developmental Studies Hybridoma Bank | Troma 1, RRID:AB_2891089 | |
| Antibody | Epcam | Biolegend | Cat #118202, RRID:AB_1089027 | |
| Antibody | GFP | Aveslabs | Cat #GFP-1020 | |
| Antibody | Sheep antidigoxigenin Fab fragments | Roche | Cat #11093274910, RRID:AB_514497 | AP-conjugated |
| Recombinant DNA reagent | T-Str-promoter | *Clements et al., 1996* Mech. Dev. 56, 139–149 | | Primitive streak specific promoter from Tbxt |
| Recombinant DNA reagent | creERT | *Jurberg et al., 2013* Dev. Cell 25, 451–462 | | Tamoxifen-inducible cre recombinase |
| Sequence-based reagent | Oligonucleotides | *Table 1* | | |
| Commercial assay or kit | DIG RNA Labeling Mix | Roche | Cat #11277073910 | |
| Commercial assay or kit | NBT/BCIP solution | Roche | Cat #11681451001 | |
| Commercial assay or kit | Blocking reagent | Roche | Cat #11096176001 | |
| Chemical compound, drug | CellTracker CM-DiI | Life Technologies | Cat #C7000 | |
| Chemical compound, drug | Proteinase K | Roche | Cat #3115801001 | |
| Chemical compound, drug | Tamoxifen | Sigma | Cat #T5648 | |
| Chemical compound, drug | RapiClear | SUNJin lab | Cat #1.49 | |

