## [Editor Report · eLife Assessment]

Morphological characteristics and phenotypes of mutations in key developmental genes suggest that head, trunk, and tail development are regulated by discernible modules. Gdf11 signalling plays a crucial role in orchestrating the transition from trunk to tail tissues in vertebrate embryos. This **important** study presents **convincing** evidence that Tgfbr1 acts upstream of Isl1 (a pivotal effector of Gdf11 signalling) and regulates blood vessels, the lateral plate mesoderm, and the endoderm associated with the trunk-to-tail transition. Together with the previous studies, this work identifies a key signal that acts as the pivot of the trunk-to-tail transition.

---

## [Referee Report · Joint Public Review]

Previously, this group showed that Tgfbr1 regulates the reorganization of the epiblast and primitive streak into the chordo-neural hinge and tailbud during the trunk-to-tail transition. Gdf11 signaling plays a crucial role in orchestrating the transition from trunk to tail tissues in vertebrate embryos, including the reallocation of axial progenitors into the tailbud and Tgfbr1 plays a key role in mediating its signaling activity. Progenitors that contribute to the extension of the neural tube and paraxial mesoderm into the tail are located in this region. In this work, the authors show that Tgfbr1 also regulates the reorganization of the posterior primitive streak/base of allantois and the endoderm as well.

By analyzing the morphological phenotypes and marker gene expression in Tgfbr1 mutant mouse embryos, they show that it regulates the merger of somatic and splanchnic layers of the lateral plate mesoderm, the posterior streak derivative. They also present evidence suggesting that Tgfbr1 acts upstream of Isl1 (key effector of Gdf11 signaling for controlling differentiation of lateral mesoderm progenitors) and regulates the remodelling of the major blood vessels, the lateral plate mesoderm and endoderm associated with the trunk-to-tail transition. Through a detailed phenotypic analysis, the authors observed that, similarly to Isl1 mutants, the lack of Tgfbr1 in mouse embryos hinders the activation of hindlimb and external genitalia maker genes and results in a failure of lateral plate mesoderm layers to converge during tail development. As a result, they interpret that ventral lateral mesoderm, which generates the peri cloacal mesenchyme and genital tuberculum, fails to specify.

They also show defects in the morphogenesis of the dorsal aorta at the trunk/tail juncture, resulting in an aberrant embryonic/extraembryonic vascular connection. Endoderm reorganization defects following abnormal morphogenesis of the gut tube in the Tgfbr1 mutants cause failure of tailgut formation and cloacal enlargement. Thus, Tgfbr1 activity regulates the morphogenesis of the trunk/tail junction and the morphogenetic switch in all germ layers required for continuing post-anal tail development. Taken together with the previous studies, this work places Gdf11/8 - Tgfbr1 signaling at the pivot of trunk-to-tail transition and the authors speculate that critical signaling through Tgfbr1 occurs in the posterior-most part of the caudal epiblast, close to the allantois.

The data shown is solid with excellent embryology/developmental biology. This work demonstrates meticulous execution and is presented in a comprehensive and coherent manner. Although not completely novel, the results/conclusions add to the known function of Gdf11 signaling during the trunk-to-tail transition.

---

## [Author Response]

The following is the authors’ response to the original reviews.

**Joint Public Review:**
Previously, this group showed that Tgfbr1 regulates the reorganization of the epiblast and primitive streak into the chordo-neural hinge and tailbud during the trunk-to-tail transition. Gdf11 signaling plays a crucial role in orchestrating the transition from trunk to tail tissues in vertebrate embryos, including the reallocation of axial progenitors into the tailbud and Tgfbr1 plays a key role in mediating its signaling activity. Progenitors that contribute to the extension of the neural tube and paraxial mesoderm into the tail are located in this region. In this work, the authors show that Tgfbr1 also regulates the reorganization of the posterior primitive streak/base of allantois and the endoderm as well.By analyzing the morphological phenotypes and marker gene expression in Tgfbr1 mutant mouse embryos, they show that it regulates the merger of somatic and splanchnic layers of the lateral plate mesoderm, the posterior streak derivative. They also present evidence suggesting that Tgfbr1 acts upstream of Isl1 (key effector of Gdf11 signaling for controlling differentiation of lateral mesoderm progenitors) and regulates the remodelling of the major blood vessels, the lateral plate mesoderm and endoderm associated with the trunk-to-tail transition. Through a detailed phenotypic analysis, the authors observed that, similarly to Isl1 mutants, the lack of Tgfbr1 in mouse embryos hinders the activation of hindlimb and external genitalia maker genes and results in a failure of lateral plate mesoderm layers to converge during tail development. As a result, they interpret that ventral lateral mesoderm, which generates the peri cloacal mesenchyme and genital tuberculum, fails to specify.They also show defects in the morphogenesis of the dorsal aorta at the trunk/tail juncture, resulting in an aberrant embryonic/extraembryonic vascular connection. Endoderm reorganization defects following abnormal morphogenesis of the gut tube in the Tgfbr1 mutants cause failure of tailgut formation and cloacal enlargement. Thus, Tgfbr1 activity regulates the morphogenesis of the trunk/tail junction and the morphogenetic switch in all germ layers required for continuing post-anal tail development. Taken together with the previous studies, this work places Gdf11/8 - Tgfbr1 signaling at the pivot of trunk-to-tail transition and the authors speculate that critical signaling through Tgfbr1 occurs in the posterior-most part of the caudal epiblast, close to the allantois.Strengths:The data shown is solid with excellent embryology/developmental biology. This work demonstrates meticulous execution and is presented in a comprehensive and coherent manner. Although not completely novel, the results/conclusions add to the known function of Gdf11 signaling during the trunk-to-tail transition.Weaknesses:The authors rely on the expression of a small number of key regulatory genes to interpret the developmental defects. The alternative possibilities remain to be ruled out thoroughly. The manuscript is also quite descriptive and would benefit from more focused highlighting of the novelty regarding the absence of Tgfbr1 in the mouse embryo. They should also strengthen some of their conclusions with more details in the results.

Although we used a limited number of key regulatory genes to interpret the phenotype, these genes were carefully chosen to focus on specific processes involving the lateral mesoderm, its derivatives, and the endoderm. In addition to these markers, we included references to other relevant markers that were previously analyzed and initially led us to examine the lateral plate mesoderm and tail gut in Tgfbr1 mutants. To strengthen our analysis, we have now incorporated additional data to clarify specific phenotypes. For instance, in situ hybridization (ISH) for Shh further confirms abnormalities at the caudal end of the endoderm in mutant embryos, while no endodermal defects are observed in the trunk region. We also included an analysis of the intermediate mesoderm, which shows abnormalities at the same level as those found in the lateral plate mesoderm and endoderm of Tgfbr1 mutants.

It’s important to note that using additional markers to assess the epiblast/primitive streak of Tgfbr1 mutants at E7.5–E8.5, as suggested by a reviewer, is unlikely to yield new insights. At these early stages, Tgfbr1 mutant embryos do not display observable phenotypes in the main body axis. Data in this manuscript already demonstrate the absence of abnormalities at this stage, as shown in Figure 3 and Supplementary Figure 6. Additionally, the expression of certain genes showing abnormalities when the embryo would enter tail development, in the trunk their expression remains unaffected, indicating that trunk extension is not significantly impacted by Tgfbr1 deficiency. While transcriptomic analysis of these Tgfbr1 mutants could provide interesting insights, it would be more appropriate to focus on later developmental stages, which would be beyond the scope of the current study.

The second major critique was that the manuscript is primarily descriptive. We disagree with this assessment. Several hypotheses were rigorously tested using genetic approaches, including Isl1 knockout experiments, cell tracing from the primitive streak with a newly generated Cre driver to activate a reporter from the ROSA26 locus, and assessment of extraembryonic endoderm fate in Tgfbr1 mutants by introducing the Afp-GFP transgene into the Tgfbr1 mutant background. Additionally, we conducted tracing analyses of tail bud cell contributions to the tail gut via DiI injection and embryo incubation. To address potential concerns regarding this experiment, we have included data showing the DiI position immediately after injection to confirm that it does not contact the tail gut. We also considered and accounted for potential DiI leakage into neuromesodermal progenitors to clarify the endodermal results.

Our genetic and DiI experiments were specifically designed to differentiate between alternative hypotheses and to confirm hypotheses generated from other analyses. Additionally, improvements in some of the imaging data have helped address remaining concerns.

**Reviewer #1 (Recommendations For The Authors):**
I have listed my suggestions as queries. The authors may perform experiments or clarify by editing the text to address them.

The authors state on Page 11 and elsewhere that the ventral lateral mesoderm is absent in the Tgfbr1 mutant. What is the basis for this conclusion? Are there specific markers for PCM or GT primordium?

The specific marker of PCM and GT primordium is Isl1. The absence of this marker in the Tgfbr1 mutants is shown in (Dias et al, 2020). The reference is introduced in the manuscript.

A schematic illustrating the VLM and the expression patterns of Tgfbr1, Gdf11, etc., would be helpful.

Characterization of Gdf11 expression has been previously reported (e.g. McPherron et al 1999, cited in our manuscript). It is expressed in the region containing of axial progenitors before the trunk to tail transition and not expressed in the VLM. As for Tgfbr1 expression is hard to detect, likely because it is ubiquitously expressed at low level. We include in this document some pictures of an ISH, including a control using the Tgfbr1 mutants to illustrate that the staining resembling background actually represents Tgfbr1 expression. If the reviewers find it important, we can also incorporate these data into the manuscript. Under these circumstances, we feel that a schematic might not be very informative.

**Author response image 1. sa2fig1:** Image showing an example of an ISH procedure with a probe against Tgfbr1, showing widespread and low expression. The lower picture shows a ventral view of a stained wild type E10.5 embryo.

Foxf1+ cells in the 'extended LPM' of Tgfbr1 mutants suggest fate transformation, or does it indicate the misexpression of marker gene otherwise suppressed by Tgfbr1 activity? The authors suggest that Foxf1+ cells are VLM progenitors from posterior PS trapped in the extended LPM. Do they continue to express PS markers?

The observation that both in wild type and Tgfbr1 mutant embryos Foxf1 expression in the trunk is restricted to the splanchnic LPM indicates that the absence of this marker in the somatic LPM is not the result of a suppression of its expression by Tgfbr1. In wild type embryos Foxf1 is also expressed in the posterior PS, regulated independently of its expression in the LPM (i.e. Shh-independent) and later in the pericloacal mesoderm (our supplementary figure 2). As Foxf1 expression in the posterior PS was not suppressed in the Tgfbr1 mutants, together with the absence of pericloacal mesoderm, we interpret that the Foxf1-positive cells in the two layers around the extended celomic cavity in the posterior end of the mutant embryos derived from the posterior PS, resulting from the absence of its normal progression through the embryonic tissues.

We did not find expression of PS markers giving rise to paraxial mesoderm, like Tbxt, further suggesting that those cells could derive from the restricted set of cells within the posterior PS that contribute to the pericloacal mesoderm

For example, the misexpression of Apela is interpreted as mis-localized endoderm cells. They show scattered Keratin 8 misexpression to support the interpretation. It would be more convincing if the authors tested the expression of other endoderm markers.

As indicated in the manuscript, we suggest that these cells are endoderm progenitors (p. 13), like those present at the posterior end of the gut tube at E9.5 and E10.5, that are unable to incorporate into the gut tube. Apela is not a general endodermal marker: it is expressed in the foregut pocket and the nascent cells of the hindgut/tail gut, becoming down regulated as cells take typical endodermal signatures. The presence of ectopic Apela expression in the extended LPM of the mutant embryos might indeed indicate the presence of progenitors that failed to downregulate Apela resulting from the lack differentiation-associated downregulation. This would also implicate the absence of definitive endodermal markers.

The Nodal signaling pathway in the anterior PS drives endoderm development. It acts through Alk7. Does Tgfbr1 (Alk5) mutation impact endoderm development, in general? It isn't easy to assess this from the Foxa2 in situ RNA hybridization shown in Figures 6A and B. It would be helpful for the readers if the authors clarified this point.

In the pictures shown in Figure 7D-D’ it is already shown that the endoderm is mostly preserved until the region of the trunk to tail transition. The presence of a rather normal endoderm in the embryonic trunk can also be seen with Shh, a figure added as Supplementary Fig.5.

**Reviewer #2 (Recommendations For The Authors):**
The authors mention two interesting novel points which they should develop in the discussion, and probably also in the results.(1) The authors speculate about the possible involvement of the posterior PS as a mediator of Gdf11/Tgfbr1 signaling activity. However, as mentioned in the manuscript, their experiments do not allow regional sublocalization within the PS... Here it would be important to assess/discuss in more detail which progenitors respond to this signaling activity and when they do it. At the very least, the authors should provide high-resolution spatiotemporal data of the expression of Tgfbr1 in the PS.

Tgfbr1 expression at this embryonic stage does not give clear differential patterns. The data reported for this expression in Andersson et al 2006 is very low quality and we have not been able to reproduce the reported pattern. On the contrary, all our efforts over the years provided a very general staining that could even be interpreted as background. When we now included Tgfbr1 mutants as controls, it became clear that the ubiquitous and low level signal observed in wild type embryos indeed represent Tgfbr1 expression pattern: low level and ubiquitous. We are attaching a figure to this document illustrating these observations. If required, this can also be included in the manuscript as a supplementary figure.

Also, the work of Wymeersch et al., 2019 regarding the lateral plate mesoderm progenitors (LPMPs) should be referred to and discussed here.

This was now added in the results (page 11) and in discussion (page 16).

For instance, are the LPMP transcriptomic differences detected between E7.5 and E8.5 caused by Tgfbr1 signaling activity? This question could be easily answered through a comparative bulk RNAseq analysis of the posterior-most region of the PS of mutant and WT embryos. The possible colocalization of Tgfb1 (Wymeersch et al., 2019) and Tgfbr1 in the LPMPs should also be addressed.

We agree with the suggestion that RNA-seq in the posterior PS of WT and mutant embryos might be informative. However, it is very likely that within the proposed timeframe (E7.5 to E8.5) that there are no significant differences between the wild type and the Tgfbr1 mutant embryos because there is no apparent axial phenotype in Tgfbr1 mutant embryos before the trunk to tail transition. Therefore, at this stage, we think that this experiment is out of the scope of the present manuscript.

(2) The activity of Tgfbr1 during the trunk-to-tail transition is critical for the development of tail endodermal tissues. Here the authors suggest again the involvement of the posterior PS/allantois region, but a similar phenotype can also be observed for instance in the absence of Snai1 in the caudal epiblast (Dias et al., 2020)... It would be important to assess/discuss the origin of those morphogenetic problems in the gut. Is it due to the reallocation of NMC cells into the CNH? The tailbud-EMT process? LPMPs specification?... Regional mutations or gain of functions of Snai1 or Tgfbr1 in the caudal epiblast would help answer the question.

The endodermal phenotype in the Snai1 mutants is different to that observed in the Tgfbr1 mutants. As can be observed in Figures 3, 4 and 5 of Dias et al. the absence of tailbud is replaced by a structure that extends the epiblast. As a consequence, the endoderm finishes at the base of that structure, even expanding to make a structure resembling the cloaca, which is different to what is seen in the Tgfbr1 mutants. In this case, the lack of tail gut is likely to result either from the lack of formation of the progenitors of the gut endoderm or from the dissociation of what would be the tail bud from the LPM. Actually, hindlimb/pericloacal mesoderm markers, like Tbx4, are preserved in the Snai1 mutant. As for the gain of function of Snai1 experiment, already reported also in Dias et al 2020, the destiny of these cells is not clear. The ISH for Foxa2 showed extra signals but as it is not an exclusive marker for endoderm it is not possible to know whether any of these signals correspond to endodermal tissues.

Regarding the development of tail endodermal tissues, the authors suggest that it occurs from a structure derived from the PS that is located posteriorly, in the tailbud, after the tip of the growing gut. This is an important and novel point as it suggests that the primordia of the endoderm is not wholly specified during gastrulation. So the observation should be well supported. How can Anastasiia et al. distinguish such "structure" from the actual developing gut? Does it have a distinct molecular signature or any morphological landmark that enables its separation from the actual gut? The data suggests that the region highlighted in Supplementary Figure 4Ab contains part of the actual gut tube (the same is suggested in Figure 5B). If the authors think otherwise, they must characterize that region of the tailbud by doing a thorough morphological and gene/protein expression analysis and assess its potency, via transplantation experiments. Also, the authors' claim mostly relies on the DiI experiments and those have three problems: #1 Anastasiia et al. assess "tail" endodermal growth at E9.5 when the correct stage to do it is after E10.5 (after tailbud formation). 2# Incongruencies, low number (only three embryos), and diversity in the results shown in Figure 8 and Supplementary Figure 4. For instance, despite similar staining at 0h, the extension and amount of DiI present in the gut tube after 20h varies significantly amongst the differently labeled embryos. A possible explanation lies in the abnormal leakiness of the DiI labelings and that is confirmed by the observations shown in Supplementary Figure 4M-O; the same for Supplementary Figure 4G, which shows a substantial amount of DiI in the neural tube. 3# The authors must provide high-quality data showing which tissues/regions were labelled at time 0h, including transversal and sagittal sections as they did for the 20h time-point. Additionally, it is important to re-orient the sagittal optical sections to a position that also shows the neural tube (like a mid-sagittal section) and include information concerning the AP/DV axis, as well as the location of the transversal optical sections in the sagittal image.

As described in the reply to reviewer 1, Apela is expressed in the nascent tail gut endoderm but not in more anterior areas except for a foregut pocket, and becomes downregulated as the tube acquires endodermal signatures. Therefore, the structure to which the reviewer refers to might indeed represent a group of progenitors that extend the tail gut. And the observation that this property is observed only in the tail gut as it grows, already separates this region of the gut, which in the end do not contribute to mature organs, from more anterior areas of the endoderm (essentially anterior to the cloaca) that will become a relevant tissue of the intestinal organs. Our DiI labelling experiment was aimed to test whether this pool of cells contributes to the gut but does not allow to determine the nature of those cells, a question that will require further research (discussed on p. 17) and we think is beyond the scope of the present manuscript.

Regarding the labelling at E10.5, we agree that the tail bud in terms of NMCs is not completely formed, for example, at E9.5 the neuropore is not yet closed. However, we are more interested in regression of the epiblast, which is complete by E9.5. Injecting at E9.5 also has technical advantages for us, first, because in our hands earlier embryos grow better in culture, and second, because it is easier to inject in the tailbud at E9.5 because it is a little bit bigger than at E10.5. Therefore, injecting at E9.5 is less prone to technical artifacts due to injection inaccuracy and compromised growth in culture.

We agree that the injected DiI could also leak into NMPs, which might be located in the same area. However, while this could result in labeling of the neural tube, it would not affect the interpretation of the finding of labeled cells in the tail gut. Indeed, the presence of this label in the gut epithelium indicates the presence of progenitors in the injected region of the tail gut. We added some considerations of this the possible leakage into the results section of the manuscript (p. 15). We thank the reviewer for drawing our attention to this issue.

We also now provide high quality data showing labelled tissue at 0h in Supplementary figure 8A-c’, higher magnification images in Fig. 8, and reoriented optical sections in Fig.6 and in Supplementary Fig. 7, including axis and location of the sections as suggested by the reviewer.

Minor concerns/comments:(1) The abstract is quite long, though this might be fine for this journal.(2) In relation to the comment on the abstract, the manuscript needs an initial Figure descrbing the events that are described in the introduction. Otherwise, the manuscript will only be accessible to mouse embryologists.

We have a figure summarizing the results at the end of the manuscript, we think that including similar figure in the beginning might be redundant. What we could do, if required, is to include this type of schematic as a graphical abstract.

(3) The authors need to clarify what they mean when they use the following expressions "PS fate" and "fate of the posterior PS".

I do not think that we have used such expressions. Indeed, they did not come out when we run a “find” in the word document. However, they would mean the tissue that would come out from them at later developmental stages.

(4) The assessment of Isl1 expression in Tgfbr1 mutant and transgenic mouse embryos would be better indicative of their molecular relationship than a comparative phenotypic analysis.

These data have been reported in Dias et al 2020 and Jurberg et al 2013, both cited in the manuscript.

(5) The authors should explain or discuss what the upregulation of Foxa2 in the posterior end of Tgfbr1 mutants means.

While an upregulation is apparent in the figure, looking at other pictures we cannot be sure of this being a significantly quantifiable up-regulation. We therefore removed the statement from the text.

(6) What happens to the intermediate mesoderm during the trunk-to-tail transition? Is Tgfbr1 involved in the regulation of its development?

We have tested this using Pax2 and added the relevant data in Supplementary Fig. 1 and described in the results.

(7) The term "potential" should not be used during the description of DiI labeling experiments as this technique only assesses cell fate.

Corrected

(8) Some figures lack AP/DV axis information (e.g. Figures 6, C, and D).

Corrected